# Technology Readiness and Economic Benefits of Swappable Battery Standard: Its Implication for Open Innovation

**Era Febriana Aqidawati [1], Wahyudi Sutopo [2,*], Eko Pujiyanto [1], Muhammad Hisjam [3], Fakhrina Fahma [3] and Azanizawati Ma'aram [4]**

1   Master Program of Industrial Engineering Department, Faculty of Engineering, Universitas Sebelas Maret, Surakarta 57126, Indonesia; erafebrianaaqidawati@student.uns.ac.id (E.F.A.); ekopujiyanto@ft.uns.ac.id (E.P.)
2   Centre of Excellence for Electrical Energy Storage Technology, Universitas Sebelas Maret, Surakarta 57126, Indonesia
3   Research Group Industrial Engineering and Techno-Economic, Department of Industrial Engineering, Faculty of Engineering, Universitas Sebelas Maret, Surakarta 57126, Indonesia; hisjam@staff.uns.ac.id (M.H.); fakhrinafahma@staff.uns.ac.id (F.F.)
4   School of Mechanical Engineering, Faculty of Engineering, Universiti Teknologi Malaysia, Johor Baru 81310, Malaysia; niza@utm.my
*   Correspondence: wahyudisutopo@staff.uns.ac.id

**Abstract:** The innovation of electric motorcycle swap-battery (EMSB) technology encourages the formation of a new ecosystem at the early stage of the supply chain. The EMSB technology has allowed an open innovation system with collaboration between supply-chain players, universities, and the government for finding a thriving solution to enable the faster adoption and diffusion of EMSB in Indonesia. Standardization is seen as a way to accelerate the downstream EMSB technology innovation to leverage the economic benefits and to support the growth of a green economy in Indonesia. This study aimed to propose a model with which to measure the technology readiness of the EMSB's stakeholders in implementing the swappable battery (SB) standard. We developed the technometric framework and Economic Benefits of Standards—ISO Methodology 2.0. We generated 13 criteria of technoware, humanware, inforware, and orgaware and 21 indicators of the standard's impacts to be utilized as a measurement model. We interviewed 11 respondents consisting of the standard regulator, research and development center, manufacturer, testing labs, and product-certification bodies. The results show the technology contribution coefficient (TCC) and earnings before interest and taxes (EBIT) value representing the SB standard's feasibility. The proposed model can evaluate the weak points and propose strategies to leverage the SB standard's technology readiness and economic benefits.

**Keywords:** technology readiness; economic benefits; swappable battery standard; open innovation; electric motorcycle swap-battery; green economy

## 1. Introduction

The penetration of electric vehicles (EV) worldwide continues to progress rapidly due to technology improvements, urban air quality concerns, and rising consumer awareness. The EV market has grown significantly in many countries. Several experts and institutions predicted that the trend would continue in the coming years, accelerating the development of EV batteries. According to several studies, the increase in EV sales since 2009 is primarily due to government policy support during the early stages of EV market penetration [1–4]. Worldwide progress in EV development and utilization varies. The world's leading EV markets include the United States, Norway, China, Japan, and the United Kingdom. Taiwan, on the other hand, has led the development of innovative electric scooter and battery-swapping systems. Meanwhile, Southeast Asian countries, e.g., Thailand, Malaysia, and Indonesia, are getting started with EV development [5–8].

The government of Indonesia has considered electric vehicles to replace fossil-fuel vehicles, marked by regulations regarding the acceleration of the battery-based electric-motor-vehicle program for road transportation. This policy is an effort to reduce the use of petroleum products so that the environment is maintained and a green economy can be achieved [5–9]. There are various methods for charging electric vehicles. However, battery-swap technology is considered a viable innovation for alternative fueling options due to its low carbon emissions, resource efficiency, socially inclusive nature, and hassle-free nature [10–14]. Therefore, it is expected that the ecosystem of electric vehicles will continue to emerge and develop in the country [15–17].

The innovation of the electric motorcycle swap-battery (EMSB) technology encourages the formation of a new ecosystem at the early stage of the supply chain, including technopreneur startups, from manufacturers, suppliers, and distributors for commercialization [18–22]. The swappable batteries (SB), electric motorcycle (EM), and battery swap/charging station (BSCS) are vital components of EMSB that have attracted attention and allowed an open innovation system with the collaboration of many parties, including supply-chain players, universities, and government, for finding a thriving solution to enable the faster adoption and diffusion of EMSB in Indonesia [23–25]. The EMSB, as a technological innovation output of research and development, must be maximally utilized to encourage economic development through downstream processes and commercialization [26,27].

Without standardization, innovation will not happen because standardization lays the groundwork for future innovation [28]. In contrast to popular belief, innovations and standardization can coexist since standards facilitate technological change, process improvement, and technology transfer across sectors and borders [29,30]. Standardization is an overall effort to ensure that standards are formed and correctly implemented, involving many parties. Like open innovation processes, the development of standards brings together knowledge and experience from various stakeholders, generating solutions that are relevant and accessible to the general public [31]. Thus, the standardization process can be characterized as a contributor to encouraging open innovation to support the development and dissemination of new technologies and the creation of new markets.

On the other hand, technology commercialization is not possible if the technology is not ready. Various challenges and problems could cause the failure of technological innovations to enter the market, which results in the technology readiness level (TRL) of the developed innovations failing to meet the criteria of the industrial market [32–34]. Therefore, TRL is critical to measure the commercialization readiness of the technology and determine the scheme and mechanism. This article discusses the relationship between standards, open innovation, and TRL. In the context of the problem of implementation, the Swappable Battery Standard for EM becomes a reference for the characteristics of the problem that can be used to describe this relationship. This is evidenced by a case study in Indonesia showing that it is impossible to innovate swap-battery technology without considering standards and TRL. Previous studies have proven the need for the relationship between innovation, standardization, and commercialization, namely, [35–40]. The open innovation approach needs to be chosen to develop standards by market needs so that standardization can support business growth in EMSB innovation. Therefore, developing a model for measuring technology readiness, open innovation, and the economic benefits of swappable battery standards for electric motorcycles in Indonesia is necessary.

The EMSB technological innovation needs to be supported as quickly as possible with standardization to accelerate the diffusion of the innovation to facilitate commercialization [40–44]. This is because standardization is supposed to facilitate technology transfer by offering privileged access to interdisciplinary know-how [45]. Additionally, standardization is expected to be a valuable tool for reducing uncertainties in technology transfer and promoting product and process development [46]. In addition, standardization is needed to contribute to trust in technology and product innovation by reducing various types of risks for both users and society, including health, safety, and environmental risks [47]. Standards

reduce the time to market for innovative inventions and technologies and enable early marketing, e.g., gathering support from all the relevant stakeholders. The development of standards can help emerging technology ecosystems to overcome problems, thereby aiding the commercialization of new products [48]. Thus, it is hoped that the EMSB stakeholders in Indonesia can immediately implement the SB standard. Therefore, it is necessary to evaluate EMSB stakeholders' readiness to adopt and enforce the SB standard to accelerate the technology commercialization process.

The governance of technological innovation must be implemented to strengthen the national capacity for technological innovation, including supporting the application of standards [49]. The measurement of technology components is one of the critical pillars in supporting technological developments. The components in question are technoware, humanware, inforware, and orgaware (THIO). These four components represent the readiness of EMSB stakeholders in Indonesia to implement the SB standard. A swappable battery standard has been established, referring to IEC 62840-2: 2016 regarding the safety requirements for electric vehicle battery-swap systems for a standardized battery pack product installed on various brands of electric motorcycles. The testing parameters in the standard include protection against electric shocks, equipment constructional requirements, electromagnetic compatibility, and marking and instructions [50]. In standardization, it is critical to measure the benefits of implementing standards and their impact on the stakeholders who implement them [51–53]. The measurement of the benefits of standards is essential for prioritizing standardization activities, increasing awareness, improving communication, promoting standards, and encouraging stakeholder participation in standardization activities [54]. From various studies and case studies that have been carried out, it is known that the application of standards will provide benefits in the form of adding economic value to the company [55]. Thus, it is crucial to study and identify the SB standard's economic benefits. SB products can be certified with a conformity-assessment body that consists of a testing laboratory and a product-certification body that certifies that the product is suitable for consumer use. Therefore, a measurement model of SB is needed that considers the readiness of battery stakeholders to implement the standard and the economic benefits of implementing the standard. Thus, it is hoped that the measurement model can be used as a foundation and contribute to the final output of a series of standardization activities, namely, the implementation of standards by battery-swap stakeholders.

Previous studies have evaluated readiness in adopting technology within the organization's scope, within the industry, and nationally. Several studies have produced models to measure the level of technology-adoption readiness according to the needs of each research and case study. Some studies have considered THIO's four technology components [56–60], and some have only considered some of the technology components [61–67]. Meanwhile, the literature on implementing standards and their economic benefits are also growing [51,52]. Researchers have assessed the economic benefits of standards by employing ISO methodology 2.0 with case studies on various standards [54,68–73]. Based on the literature review that has been carried out, there has been no research that has measured technology readiness for adopting standards while, at the same time, assessing their economic benefits, and vice versa. Therefore, a model for measuring technology readiness for adopting a standard and, at the same time, assessing the economic benefits of implementing the standard is needed.

This study aimed to design a model that considered THIO and the economic benefits of a standard to measure the technology readiness and feasibility of EMSB stakeholders for implementing the SB standard. The expected outputs are a description of the readiness of EMSB stakeholders in Indonesia and the economic benefits of implementing the SB standard, an overview of investments before the standard is implemented, and proposed recommendations to EMSB stakeholders. It is hoped that this research will be able to be used as an initial foundation before implementing the standard and help to accelerate the program for battery-based electric motor vehicles through study of the implementation of the swappable battery standard.

## 2. Materials and Methods

*2.1. Defining the Structural Framework*

The design of the model for measuring technology readiness and the economic benefits began with the operationalization of the variables that would be used as a measuring tool in assessing the readiness, open innovation, and economic benefits of implementing the SB standard. At this stage, we explained the variables to be studied and described, in detail, the criteria and indicators considered. These variables came from selected concepts that were arranged in the framework of thinking. This section is divided into two, namely, the operationalization of variables for measuring the readiness of stakeholders for implementing the battery-swap standard and the operationalization of variables for assessing the open innovation and economic benefits of implementing the standard.

We conducted a preliminary study and generated an initial framework to develop the model [74]. In this study, we developed a model to measure the readiness of EMSB stakeholders based on a technometric approach that considers four technology components: technoware, humanware, inforware, and orgaware [49]. Technoware is object-embodied technology, physical facilities, or technical equipment, which refers to the physical capital used to perform various jobs carried out by all organizations. Humanware is person-embodied technology and human abilities, which refers to anything that makes someone at work do something. This manifests in what that person does with the technology available by applying personal qualifications and experience. Inforware is document-embodied technology, document facts, and information tools, which refer to codified technical knowledge related to specific work requirements and work conventions that provide the basis for any technology system used in work performed by different organizations. Meanwhile, orgaware is institution-embodied technology, organizational frameworks, organizational/institutional tools, and regulations based on the logic of system integration and the coordination of activities and resources to achieve the planned goals of a particular job. Each of these components has different criteria and was determined based on previous literature studies while considering the operational definition of the component. After the criteria were determined, the next step was to translate them into indicators that would be assessed in the questionnaire. The results of identifying the criteria and indicators for each component of readiness can be seen in Table 1.

Meanwhile, the measurement model for assessing the economic benefits of implementing the SB standard was adapted from the ISO Methodological 2.0 concept. This methodology can be applied to all companies and industry sectors to identify the contribution made by the standards to performance. The primary purpose of this method is to quantitatively assess the contribution of standards to value creation in an organization. The benefits of standards can be identified along the company's overall value chain using this method [55]. In this study, the benefit considered was the impact of the standard's application on the battery-manufacturing industry. Therefore, the concept of the ISO methodology is seen as an appropriate principle for assessing the economic benefits of SB standard implementation. The operationalization of the economic benefits assessment variable was prepared based on the principles of the ISO methodology to identify the impacts arising from the application of the standard. The contribution or impact of the standard on every business function of the company was identified. The identified impacts can be quantified, and their value is expressed in an operational indicator, including cost and time. In addition, the identification of impacts depends on whether or not the impact is significant in contributing to the company's performance. Table 2 shows the operationalization of the variable for assessing the economic benefits of implementing the standard.

**Table 1.** Operationalization of technology readiness measurement variables.

| Technology Component | Criteria | Description | Indicator |
|---|---|---|---|
| Technoware | Infrastructure [75] | The available infrastructure can support the standard application process correctly. | Availability of the Conformity-Assessment Agency as the institution that issues conformity-assessment certificates/test results. |
| | | | The available capacity is adequate. |
| | Equipment [56,57,65,76,77] | Equipment readiness of the conformity-assessment agencies to test the swappable batteries. Testing parameters include protection against electric shocks, construction equipment, and electromagnetic compatibility. | Equipment readiness for testing protection against electric shocks in battery-swap systems, including protection against direct contact, stored energy, fault protection, and protective conductors. |
| | | | Equipment readiness to test construction equipment of battery-swap system, including mechanical-switch-device characteristics, clearances and creepage distances, and the strength of materials and parts. |
| | | | Equipment readiness for testing the electromagnetic compatibility of the battery-swap system. |
| Humanware | Human resource (HR) competence [56,60,76,78–82] | Human resources' capabilities, expertise, and skills are involved in implementing standards, standardization, and conformity assessment. | Availability of competent human resources. |
| | Human resource development [58,75,77,82,83] | Efforts to improve the quality and quantity of human resources from various stakeholders. | Increasing the number of HR for optimal roles. |
| | | | Improving the quality of human resources through training and coaching. |
| | Level of awareness [75,77,83] | The level of awareness of various parties regarding the application of standards. | Stakeholder HR awareness of the importance of implementing SB standard. |
| | Accommodation of the standard [83] | The ability of employees to accommodate the standard system. | The ability of stakeholders to adapt to the SB standard. |
| Inforware | Information systems [65] | An information system is available that can support standard implementation activities by providing the information needed by various parties. | Ease of access. |
| | | | Well-integrated system. |
| | | | Information system capabilities. |
| | Regulations/policies [75,82] | Availability of written regulations, procedures, and programs. | Regulatory effectiveness. |
| | | | Policies are well conveyed. |
| | | | Easy to understand. |
| | Communication [57,84] | Promotion and socialization to various parties related to standardization and conformity assessment. | Effectiveness of promotion and outreach programs. |
| Orgaware | The strategic plan [82] | Objectives, strategic targets, policy directions, and performance targets are the primary reference in preparing plans and implementing programs and activities carried out. | Availability of strategic plan to implement SB standard. |
| | Framework [75,79,80] | Procedures for implementing standards through appropriate conformity-assessment activities. | The efficiency of the conformity-assessment scheme. |
| | Cooperation [81,82] | Collaboration with various parties to build public awareness and interest in the application of standards. | Society participation. |
| | | | Involvement of industry and conformity-assessment bodies. |
| | Financial [57,75,78,81,83] | The government provides sufficient budget support for standard implementation. | Financial adequacy. |

**Table 2.** Operationalization of economic-benefit assessment variables.

| Business Function | Impact Indicators | Description |
|---|---|---|
| All functions | Better internal information transfer | Using standard documents and specifications makes internal information about products and services more efficient. |
| | More efficient staff training | Staff could be better trained if they had standard product specifications. |
| Management and administration, marketing and sales | Sales increase | Higher sales and increased profits due to customer confidence in standard products. |
| | Increase in average profit | |
| Management and administration, engineering, and production | More effective quality management | Quality management can be implemented more effectively based on standards. |
| | More effective HSE management | Health/safety/environmental (HSE) management failure rates are reduced due to standard implementation. |
| Engineering, R&D, and procurement | More efficient internal standardization | It is cheaper to implement standards within the company using open consensus-based standards than to develop internal standards. |
| | Product specifications are more precise | Standard supplier product specifications and customer requirements make it easy to collect relevant information. |
| Production | Improvement of production efficiency | Due to the reduced number of non-standard products, production can be more efficient. |
| Research and development | Increased efficiency of R&D activities | Standards provide free technical information, so research needs and product development costs are reduced. |
| Engineering and production | More efficient assembly | The assembly process is more efficient due to the modular product architecture. |
| Management and administration | Reduced liability costs | The cost of the obligation can be reduced if compliance with the standard is proven. |
| | Reduced operational risk | Operational risk is reduced if the product is based on standards because standard products can be sold for longer and inventories are available for longer. |
| Engineering | More efficient project development | Project development costs are reduced because the standard provides free technical information. |
| | Better quality of equipment and supplies | Higher-quality, standards-based tools and equipment can reduce failure rates and associated repair costs. |
| Inbound logistics | Better product availability | Due to the high availability of standard products, less inventory needs to be kept in the warehouse. |
| Marketing and sales | Increased competition | Lower market share because there are more competitors in the market for standard products. |
| | Reduced time to market | For products based on standards, the time to market is shorter, and market share is higher due to access to technical information and better development time. |
| | More efficient contract agreements | The defined specifications of the company's products and customer requirements make contractual agreements easier. |
| After-sales service | Product-quality improvement | Improved standard product quality means less consultation with customers is required. |
| | Better communication with customers | Can communicate information about products to customers more effectively using standard specifications. |

## 2.2. Questionnaire Design

The questionnaire designed in this study was used to obtain the level of readiness and predict the economic benefits of implementing the standard. The question items or statements in the questionnaire were prepared based on the indicators that were identified at the variable-operationalization stage. Then, each item was coded to facilitate further data processing. Based on the operationalization of the variables in Tables 1 and 2, 13 technology-readiness criteria (26 readiness indicators) and 21 standard impact indicators were obtained.

The model developed in this study used the help of a measurement instrument in the form of a questionnaire with an interval scale. The interval scale was used in this study because data collection using a ratio scale is very difficult. In determining the level of readiness for a standard application, the interval scale used was in the form of answer choices for the appropriate conditions, namely, "low", "medium", "high", "top", and "ideal". This scale refers to the research of Sharif [49] in measuring the degree of the sophistication of technological components. Meanwhile, to measure the value of the economic benefits, an interval scale was also used, but the answer choices were in the form of the estimated percentages of economic benefits obtained, namely, 0–20%, 20–40%, 40–60%, 60–80%, and >80%.

## 2.3. Questionnaire Testing

Before the questionnaire was used to collect data, the questionnaire was tested by distributing it to 11 experts/practitioners using the purposive sampling method. The test results were then used to test the validity and reliability. Question items that were invalid or unreliable were then corrected, modified or, if necessary, removed, and then, the questionnaire was re-tested.

The validity test referred to in this study is a process carried out to ensure that the statement items in the research instrument can measure the research object correctly. Validity testing is carried out by calculating the correlation between each statement item score on the relevant variable. To test the validity in this study, a non-parametric test using the Spearman correlation test was employed. The criteria used to determine whether or not a question item was valid included the item's Spearman correlation coefficient ($r_s$). If $r_s \geq 0.30$, then the question item was considered valid, and if $r_s < 0.30$, then the question item was invalid [85]. We chose 0.3 as the threshold because 0.3 becomes the cut-off point for low positive correlation. Otherwise, the correlation would be neglected.

A research instrument is reliable if it produces the same data when used several times to measure the same object. Reliable research results are data that have similarities across several research activities carried out at different times. The reliability test in this study was carried out using Cronbach's alpha formula. A questionnaire has an acceptable level of reliability if the Cronbach's alpha value > 0.70 [86–88].

## 2.4. Model Implementation Framework

The technology-readiness and economic-benefit measurement model that was designed was applied to related stakeholders by distributing questionnaires to selected experts/practitioners to represent stakeholders involved in standardization activities. The primary data-collection stage was carried out by distributing questionnaires to selected experts/practitioners to represent the stakeholders involved in standardization activities. A total of 11 respondents consisted of representatives from the national standardization body, a standard regulator; research and development teams in the field of batteries and electric vehicles from universities; and representatives from conformity-assessment agencies, namely, testing labs and product-certification bodies. The data were collected through interviews. Then, the survey results were recapitulated based on the respondents' answers, and data on the level of readiness and on the economic benefits were obtained.

Data processing was carried out to obtain a synthesis of stakeholder readiness levels based on each criterion and indicator for each technology component. These data were then further processed to calculate the value of the technology contribution coefficient (TCC),

which represents the ability of EMSB stakeholders to implement the standard. Meanwhile, data processing for the identification of the impact of standard implementation was carried out to synthesize the economic benefits that the battery industry could potentially derive, which are expressed through a measure called the EBIT (Earnings Before Interest and Tax). Then, from these two aspects, the costs component of increasing readiness and the benefits component of implementing the standard could be generalized to calculate the benefit–cost ratio so that the SB standard's feasibility could be evaluated. The framework for implementing the model is illustrated in Figure 1.

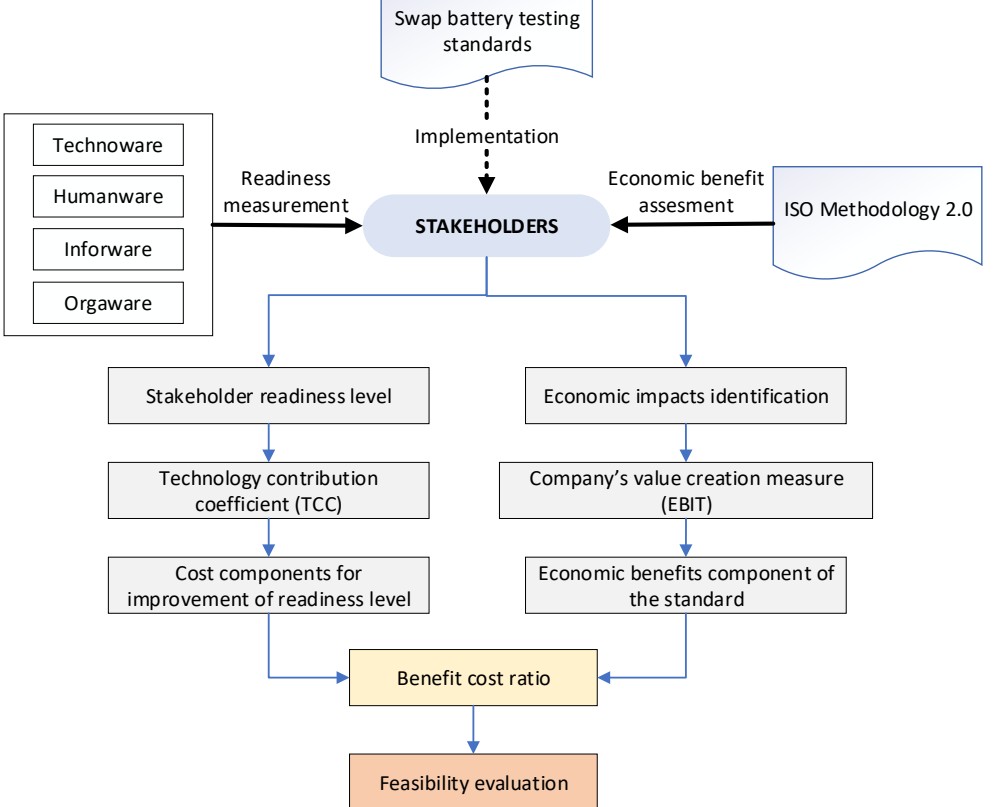

**Figure 1.** Model implementation framework.

2.4.1. Swappable Battery Standard Considering Open Innovation

This research continues previous research that has designed a national standard for swappable batteries. The open innovation approach is applied at the standard development stage, and the swap-battery technology development from the manufacturer by utilizing available external knowledge as a source of internal innovation. The open innovation framework in standards development aims to explore various stakeholders' thoughts, ideas, and needs to produce standards with technical specifications that follow domestic conditions and capabilities. The stakeholders involved include the government, R & D institutions, battery-swap laboratories, battery-swap manufacturers, electric motorcycle manufacturers, and electric motorcycle users. From the results of the analysis of the needs and capabilities of stakeholders and a comparison with the reference standard IEC 62840-2-2016, the output of the national standard for swappable battery products for electric motorcycles in Indonesia is produced [89].

2.4.2. THIO Represents the Technology-Readiness Measurement Instrument

Technology is all knowledge, products, processes, equipment, methods, and systems to create products or services [90]. Another definition of technology is the study of the manmade world, meaning that it deals with the creation or engineering of nature and solutions from and for humans themselves. According to the United Nations Economic

and Social Commission for Asia and the Pacific (UNESCAP), technology is the result of a combination of components in production that interact dynamically in a transformation process. The four basic components are technical facilities (facilities), human capabilities (abilities), information (facts), and organization (framework). Therefore, technology is the accumulation of knowledge embodied in the form of product creation, methods, processes, equipment, and services, as well as engineering, to meet the goals or expectations of human needs. According to the Technology Atlas Team and the Asian Pacific Center for the Transfer of Technology (1989), technology is composed of four components, namely, technoware, humanware, inforware, and orgaware. Figure 2 depicts the framework of technology components in technometrics (T, H, I, O) according to Smith and Sharif [91], which plays a role in creating and determining the competency position of a company:

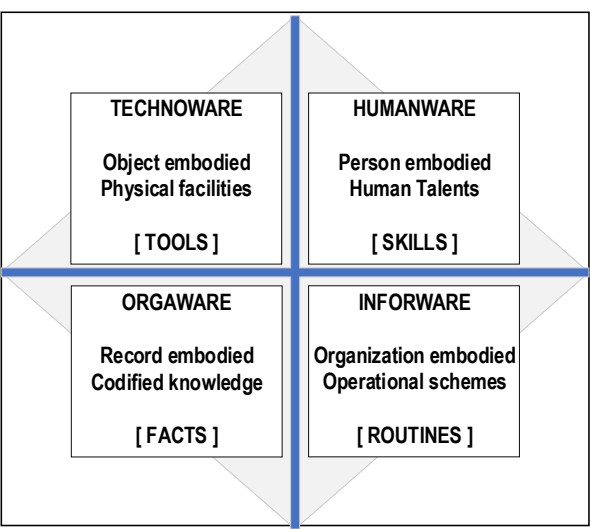

**Figure 2.** Technology system components. Source: Smith & Sharif [91].

There is a relationship between one component and another. Technoware, which is equipment and physical components in the industry, will function properly when operated by humanware components, where the operational capabilities of humans will be in line with the operational capabilities of the equipment, based on the received information components, where the flow of information will determine the movement of the humanware components and technoware, within the framework set by orgaware, where organizational policies will determine the flow of information required (inforware) by the humanware and technoware components.

Technology readiness is a systematic measurement system that supports the assessment of the maturity or readiness of a particular technology and a comparison of the maturity or readiness between different types of technology. Technology readiness can be interpreted as an indicator that shows how ready/mature a technology is to be applied and adopted by users/prospective users. Technology adoption is defined as the acceptance or use of a new technology by the adopter delivered by the technology carrier [92]. Readiness to adopt technology can be measured by integrating technometric methods with key components of technology (THIO). The technometric approach aims to measure the contribution of the technology component (THIO) in the process of transforming input into output, called the combined contribution, which is expressed using the Technology Contribution Coefficient (TCC) [93]. TCC symbolizes the ability of a system to adopt technology.

Technology Readiness Level (TRL) is a measurement system that supports the assessment of the level of maturity of a particular technology and constant comparisons between various types of technology [94]. TRL is used as a measure of the level of technology readiness, which is defined as an indicator that shows how ready or mature a technology is to be applied and adopted by users/prospective users. TRL is a systematic measurement system that supports the assessment of the maturity or readiness of a particular technology

and a comparison of the maturity or readiness between different types of technology. TRL is a measure that shows the stage or level of maturity or technological readiness on a scale of 1–9, where one level to another is interrelated and forms the basis for the next level [95].

TRL, conceptually, should contain five main considerations, namely: (a) basic research on the latest technology and concepts/methods, targeting and identifying objectives, but not relating to specific systems; (b) focused on one technology development based on specific technologies for one or more identified applications; (c) technology development and demonstration of each specific application prior to full development of the application; (d) system development (via prototyping, creation of the first unit); and (e) system launch and operation [94].

### 2.4.3. ISO Methodology

ISO has developed an approach to measuring the economic benefits of implementing a standard known as EBS (Economic Benefit of Standard). This approach is a guide issued by ISO to measure the economic impact of applying external standards [55] and can be used for both voluntary and mandatory standards. More specifically, the main purpose of this method is to quantitatively assess the standard's contribution to value creation in an organization. The benefits of standards can be identified along the company's overall value chain and its external interfaces. Benefits can be measured and expressed as impacts on certain operational indicators and converted into financial units [55]. An illustration of the EBS methodology at a glance is shown in Figure 3.

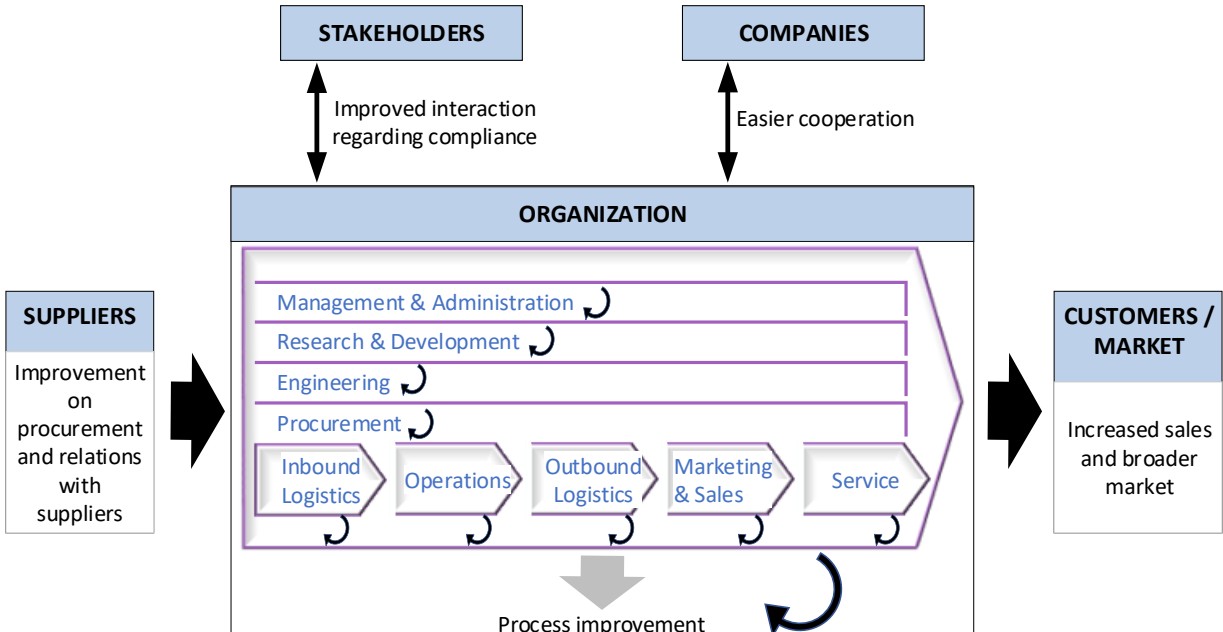

**Figure 3.** The ISO Methodology at a glance.

The main scope of the ISO methodology is to assess the economic benefits of the standard (i.e., the standard's contribution to the creation of economic value) for a company. This methodology can be applied to the economic impact of the standard on the industrial sector at a national or international level. The ISO methodology can also be adapted to describe and measure the non-economic benefits of the standard, i.e., the contribution that the standard makes to the achievement of social and environmental benefits. The methodology is focused on measuring the benefits resulting from the use of the standard. The benefits associated with participation in standards development are addressed only at the qualitative level. Several methods that can be developed to support ISO methodology are before–after comparisons, comparing concurrent conditions–projects, and what-if comparisons [69].

### 2.4.4. EMSB Stakeholders

The model that has been designed is applied through case studies on stakeholders in Indonesia. This begins with collecting data through surveys or distributing questionnaires to experts/practitioners representing each stakeholder. The selected respondents are experts, practitioners, academics, and researchers who are experts in the field of battery electric vehicles (BEV) and standardization activities. Thus, the sample of respondents in Indonesia is limited. Eleven respondents were selected to represent regulators, research and development teams (R&D) of batteries and electric vehicles from universities, and testing labs and product-certification bodies.

As the standard regulator, the national standardization body has the authority to formulate and set standards, carry out coordination in the field of standardization, and provide guidance and training to parties implementing standards. The conformity-assessment agency consists of a testing laboratory and a certification body as the party authorized to provide test-worthy certification on swap-battery products. The lithium battery manufacturer or industry (as the party that produces the battery and its derivatives), academics and researchers (as parties involved in researching and developing battery products) must refer to the standard. Then, the respondents surveyed were also part of the four stakeholders above, so they were considered able to understand things related to standardization, especially for swappable battery standards, and could answer the objectives of this research.

### 2.4.5. Technology-Readiness Measurement

In this study, technological readiness is reflected through the ability of a system to adopt technology, in this case adopting swappable battery technology and its standards. This can be measured using the Technology Contribution Coefficient (TCC). TCC is the total contribution of technology components that play a role in a system, taking into account the intensity of the contribution of each component. The Technology Contribution Coefficient (TCC) shows the technology contribution from the total transformation of inputs to outputs. By using the values of $T$, $H$, $I$, $O$, $\beta_T$, $\beta_H$, $\beta_I$, and $\beta_O$, TCC can be calculated using the equation [93]:

$$TCC = \alpha \times T^{\beta_T} \times H^{\beta_H} \times I^{\beta_I} \times O^{\beta_O} \tag{1}$$

where:

$\alpha$ = the trend factor of the technology;
$T$ = value of contribution of technoware components;
$\beta_T$ = intensity value of technoware component contribution;
$H$ = contribution value of humanware components;
$\beta_H$ = humanware component contribution intensity value;
$I$ = inforware component contribution value;
$\beta_I$ = inforware component contribution intensity value;
$O$ = contribution value of orgaware components;
$\beta_O$ = contribution intensity value of orgaware component.

TCC has several attributes, namely:

- Equation (1) implies that T,H,I,O is a non-zero function if TCC is also non-zero. This means that there is no transformation activity without the presence of the four technologies.
- To improve the state of technology through the degree of sophistication of one component, the other components are considered constant.

$$\delta\,(TCC)/\delta T = \beta_T(TCC/T) \tag{2}$$

- Overall, increasing the degree of sophistication of the four components results in the following equation:

$$\frac{dTCC}{TCC} = \beta_T \left( \frac{dT}{T} \right) + \beta_H \left( \frac{dH}{H} \right) + \beta_I \left( \frac{dI}{I} \right) + \beta_O \left( \frac{dO}{O} \right) \tag{3}$$

Equation (3) shows that the proportional increase in TCC will be equal to the sum of the proportional increases of the four components (measured by $\beta$). When the four components are increased by the same proportion ($\rho$), Equation (3) turns into the following Equation (4):

$$\frac{dTCC}{TCC} = \rho \left[ \beta_T + \beta_H + \beta_I + \beta_O \right] \tag{4}$$

If $\beta_T + \beta_H + \beta_I + \beta_O \geq 1$ or $\beta_T + \beta_H + \beta_I + \beta_O = 1$ or $\beta_T + \beta_H + \beta_I + \beta_O \leq 1$, then the TCC function is, successively, in conditions of increasing, neutral, or decreasing return to scale.

Steps in measuring TCC:

- Determine the technology component assessment matrix. This is a set of variables for each technology component, which is shown in Table 1.
- Estimate the degree of sophistication.
- In estimating the degree of sophistication, an assessment rubric is needed that becomes a reference in providing a score for each indicator. The score used refers to a study conducted by Sharif [49] in measuring the degree of sophistication of technological components. The score consists of a score of 1–5, which indicates the level of readiness of low, medium, high, top, or ideal, respectively.
- Determine the trend factor of the technology ($\alpha$) and the weight of each technology component ($\beta$).

In theory, the weight of each component can be represented based on the priority portion of the development or distribution of each technology component to each other; to facilitate benchmarking between components, it is assumed that the value of each component is the same or 0.25. While the value of is a trend factor, or scale, which in this study relates to the interests of making comparisons between industries, the value of was initially assumed to have no effect, or equal to 1; then, each component has a maximum scale of 5. So that the formula can be normalized, TCC becomes:

$$\begin{aligned} TCC &= 1 \times \frac{T^{0.25}}{5} \times \frac{H^{0.25}}{5} \times \frac{I^{0.25}}{5} \times \frac{O^{0.25}}{5} \\ &= \tfrac{1}{5} \times \left( T^{0.25} \times H^{0.25} \times I^{0.25} \times O^{0.25} \right) \\ &= 0.2 \times \left( T^{0.25} \times H^{0.25} \times I^{0.25} \times O^{0.25} \right) \\ &\quad \text{with } \alpha = 0.2 \end{aligned}$$

- Determine the upper and lower limits of each technology component.

For taking the value of each component based on the results of data collection, it can be seen that it has a range, meaning that it has an upper and lower limit, which describes the variation in conditions. For the assessment of the technology contribution coefficient (TCC), the upper limit of the available data values is used.

### 2.4.6. Economic Benefits Assessment

The steps in calculating the economic value of the application of standards in accordance with the ISO methodology are as follows:

a    Determining the Value Chain

The ISO methodology is based on a value chain approach. The value chain is a chain of activities related to a particular output, product, or service it produces. The output of the system/work has gone through the entire chain of activities that are organized and provide added value at every stage it goes through. These stages can be managed within one company, or within several different companies and these companies support each other in the supply chain network. The corporate value chain represents the chain of activities



carried out within a company. The company's operations are divided into a number of key business functions. Each of these functions corresponds to a specific set of value chain activities. The value chain of battery-swap companies was determined using the variables in Table 2.

b.    Value Drivers Analysis

Value drivers are important capabilities of a business organization that can provide a competitive advantage to the organization. The impact of standards on the company's operating processes can be assessed through the value driver or from the value creation that can be created by the company with the implementation of standards. For this reason, it is very important to consider activities that are crucial to the value of creation in order to identify whether these standards have an impact or not. If it is seen as having no impact, it can be considered as a second option. Impacts associated with implementing the standard can be selected from the standard impact map provided by ISO methodology.

a    Identification of the Impact of the Standard

After identifying the value drivers, the next step is to identify the impact of the standard. In determining the economic benefits of applying the standard, the benefits are grouped into three, namely: quantitative, semi-quantitative, and qualitative. Only quantitative benefits will be measured for their contribution to the application of standards to the object of research.

Quantitative benefits of standards are the economic benefits of implementing standards related to value drivers and can be quantified. Meanwhile, the qualitative benefits of standards are the benefits of applying standards to an object of research that are not related to the value drivers of the object of research and cannot be quantified. This qualitative impact is an intangible benefit from the application of standards to the object of research.

b.    Assessment and Consolidation of Results

The final stage in assessing the economic benefits of implementing the standard using the ISO methodology is to assess and consolidate the results of the impact of the standard. For this reason, it is necessary to identify the value of each impact of the standard on the company, then the value of each aggregated into one. In the standard impact assessment, there are two stages, namely, the selection of relevant impacts and the identification of Earnings Before Interest and Tax (EBIT) Impact. Changes in the quantitative value of the selected indicators obtained from comparisons before using the standard and after using the standard identified as the impact of implementing the standard are summed for all business functions and expressed with the financial value of EBIT (Earnings Before Interest and Taxes) as a key indicator. EBIT expresses the company's gross profit, namely, revenue minus costs. If the EBIT value is not available, then the economic impact of implementing the standard can be expressed by the number of sales available [55].

The purpose of the assessment process is to determine the impact of using standards as a measure through quantifiable indicators. This can be achieved in the final assessment through the stage of quantifying the impact of the standard into financial values. The use of standards is expected to lead to changes in the selected indicators, in various ways resulting in value creation that can be created by the company. These creations can have the impact of reducing operational costs or increasing income or through a combination of both. Depending on the operational indicators, the financial impact can be directly measured, or if data are not available, the indicator can be determined based on the estimated financial data of other similar companies.

2.4.7. Benefit–Cost Ratio and Feasibility Study

The feasibility study in this study was compiled by comparing the overall costs and benefits. The cost component is generalized based on the results of the measured readiness level. Meanwhile, the benefit component is compiled based on the identification of impacts and the estimated value of the impacts that have been carried out.

a    Costs component

The generalization of the cost components in this study refers to the cost model from [56,96], which classifies costs into three categories, namely, capital expenditure, implementation costs, and training costs.

- Capital expenditure is the cost to obtain the physical components of the technology system. In this study, capital expenditure is the cost incurred for the repair of technoware components, which involves infrastructure and equipment. This is because, based on the analysis of the level of readiness, it is known that it is necessary to make improvements to the technoware component to improve its readiness condition.
- Implementation costs are costs related to contractors, which include costs to install each piece of hardware and hourly labor costs. In this study, implementation costs are defined as costs for implementing standards, where this cost is an aggregation of all costs that must be incurred by the stakeholders involved. These types of costs are generalized based on readiness indicators for the humanware, information, and orgaware components.
- Training costs are costs associated with bringing in experts in the field of technology operation. Costs related to training costs include the cost of bringing technology experts to train employees, consisting of training materials, expert fees, and costs for employees who attend training.

Based on the explanation above, the cost components can be arranged as follows in Table 3.

**Table 3.** Costs component.

| | Readiness Indicator | Type of Cost | Notation |
|---|---|---|---|
| **Technoware** | Availability of conformity-assessment agency | Accreditation fee for additional testing scope and swap-battery system certification | $C_{T_1}$ |
| | Capacity of conformity-assessment agency | Cost of procurement of tools and machines for large capacity | $C_{T_2}$ |
| | Electric shock protection test | Cost of equipment and machinery procurement for standard testing parameters for swap-battery systemSoftware costs | $C_{T_3}$ |
| | Construction equipment test | | $C_{T_4}$ |
| | Electromagnetic compatibility test | | |
| **Human-ware** | HR quality improvement | The cost of training and human resource development for LPKs, standard regulators, and the battery industry | $C_{H_1}$ |
| **Inforware** | Standardization information system and conformity assessment | Technical Service Office service quality improvement fee | $C_{I_1}$ |
| | | Information system upgrade costs | $C_{I_2}$ |
| | Regulations/policies | Policy development costs | $C_{I_3}$ |
| | | Standard development costs | $C_{I_4}$ |
| | Effectiveness of promotion and outreach programs | Costs for counseling, workshops, seminars, and dissemination related to SNI | $C_{I_5}$ |
| **Orgaware** | Efficiency of conformity assessment scheme | Cost of developing a conformity assessment scheme for a battery-swap system | $C_{O_1}$ |
| | | Certification fee | $C_{O_2}$ |
| | | Testing fee | $C_{O_3}$ |

From the table above, the cost components can be formulated as follows:

Technoware improvement cost $(C_T) = C_{T_1} + C_{T_2} + C_{T_3} + C_{T_4}$;
humanware improvement cost $(C_H) = C_{H_1}$;
inforware improvement cost $(C_I) = C_{I_1} + C_{I_2} + C_{I_3} + C_{I_4} + C_{I_5}$;
orgaware improvement cost $(C_O) = C_{O_1} + C_{O_2} + C_{O_3}$;
Therefore, the total cost becomes: $C = C_T + C_H + C_I + C_O$.

a　　Benefits component

The generalization of the benefits component is obtained from the analysis of the benefits assessment that has been carried out, where the identification of the most significant impacts is obtained. From the analysis that has been carried out for the assessment of economic benefits, it is known that operational indicators are a measure of the value of the impact caused and expressed as an entity that can be quantified, either in the form of cost savings or increased revenues. The aggregate of all operational indicators is expressed in EBIT and represents the value of the economic benefits that will be obtained by the company. Therefore, in this case, the EBIT value is used as a benefits component to formulate the techno-economic formulation. The benefit components can be generalized from each operational indicator and are shown in Table 4.

**Table 4.** Benefit components.

| Operational Indicator | Notation |
|---|---|
| Labor cost savings per year | $B_l$ |
| R&D cost savings | $B_r$ |
| Sales increase | $B_s$ |
| Reducing costs in handling rejects, rework, and repair of defective products | $B_f$ |
| Production cost savings | $B_p$ |
| Reduction in inventory cost | $B_i$ |
| Reduction in warehousing costs | $B_w$ |
| Reduced work accident insurance costs | $B_{ai}$ |
| Waste-handling cost savings | $B_{wm}$ |
| Energy cost savings per unit production volume | $B_e$ |
| Cost reduction for warranty compensation payment | $B_{wc}$ |

From the table above, the benefits can be formulated as follows:

$$B = B_s + B_r + B_l + B_f + B_p + B_i + B_w + B_{ai} + B_{wm} + B_e + B_{wc} \tag{5}$$

Based on the generalization of techno-economic criteria in the form of cost and benefit components, a techno-economic benefit–cost ratio model can be developed. Thus, the B/C ratio formula for the implementation of standard SB is obtained, which is as follows:

$$B/C = \frac{\left(B_s + B_r + B_l + B_f + B_p + B_i + B_w + B_{ai} + B_{wm} + B_e + B_{wc}\right)(P/A, i\%, n)}{(C_T + C_H + C_I + C_O) + (O\&M)(P/A, i\%, n) - SV(P/F, i\%, n)} \tag{6}$$

where:

$PW$ = present worth;
$SV$ = salvage value = amount of assets that can be sold at the end of their useful life;
$O\&M$ = proposed project operation and maintenance costs;
$i$ = effective interest rate per interest period;
$n$ = project period (years);
$P$ = amount of money now;

*F* = future amount of money;
*A* = end of period cash flow (or equivalent period end value).

If the example of calculating the EBIT value in the previous section is applied in the B/C ratio formulation above, the formulation becomes:

$$B/C = \frac{(EBIT - tax)\,(P/A, i\%,\, n)}{(C_T + C_H + C_I + C_O) + (O\&M)(P/A, i\%,\, n) - SV(P/F, i\%,\, n)} \tag{7}$$

## 3. Results

### 3.1. Technology-Readiness Measurement and Economic-Benefit Assessment Model

Based on the operationalization of the variables and the design of the questionnaire, a model for measuring readiness and open innovation and assessing the economic benefits of standard implementation could be developed. The model is illustrated in Figure 4 The readiness measure consisted of 13 criteria. The technoware component consisted of infrastructure and equipment criteria. Meanwhile, the humanware criteria consisted of human resource (HR) competency, HR development, the level of awareness, and the accommodation of standards. The inforware criteria consisted of information systems, regulation, and communication.

Meanwhile, the orgaware criteria consisted of strategic planning, frameworks, cooperation, and financial aspects. Each criterion had indicators of readiness that stakeholders must meet. The measure of stakeholder readiness was defined as ranging from a low to an ideal level, represented by 1–5. Furthermore, the contribution of each component to the overall level of readiness can be expressed as the technology contribution coefficient (TCC), which was calculated based on the readiness value of each technology component: technoware, humanware, inforware, and orgaware. Thus, the TCC represents the technological sophistication and technological readiness to implement the SB standard.

The economic benefits assessment consisted of 21 indicators of the impacts that the battery-manufacturing industry could potentially have due to the implementation of the SB standard. The impact indicators were generalized from activities along the company's value chain. Then, from these twenty-one indicators, five impacts were chosen that were estimated to be the most significant and have the greatest impact on the battery industry. After the most significant impact was determined, the estimated impact value was assessed. The percentage change expresses the impact value of the related matters due to the implementation of the standard.

Moreover, value drivers can be analyzed based on identifying the most significant impact and on the business function where the impact provides added value. Value drivers are critical organizational capabilities that give companies a competitive advantage. Furthermore, each standard impact has operational indicators, measurable variables from company activities that show an increase or decrease in performance. Appropriate operational indicators were selected to measure the contribution to corporate value creation (EBIT). Thus, the EBIT value, which represents the company's gross profit, can be calculated from the aggregation of all these operational indicators. The EBIT value represents the economic benefits of implementing the SB standard in this study. Moreover, synergy from EMSB stakeholders is needed to support the successful implementation of the SB standard in Indonesia. The process of formulating and developing standards is the responsibility of the standard regulator as a supervisor, developer, and coordinator of activities in the field of standardization. Standard development also needs to pay attention to aspects of product development carried out by the R&D team, from both universities and related agencies. After the standard has been formulated and ratified, it will be applied to the mass production process at the battery factory.

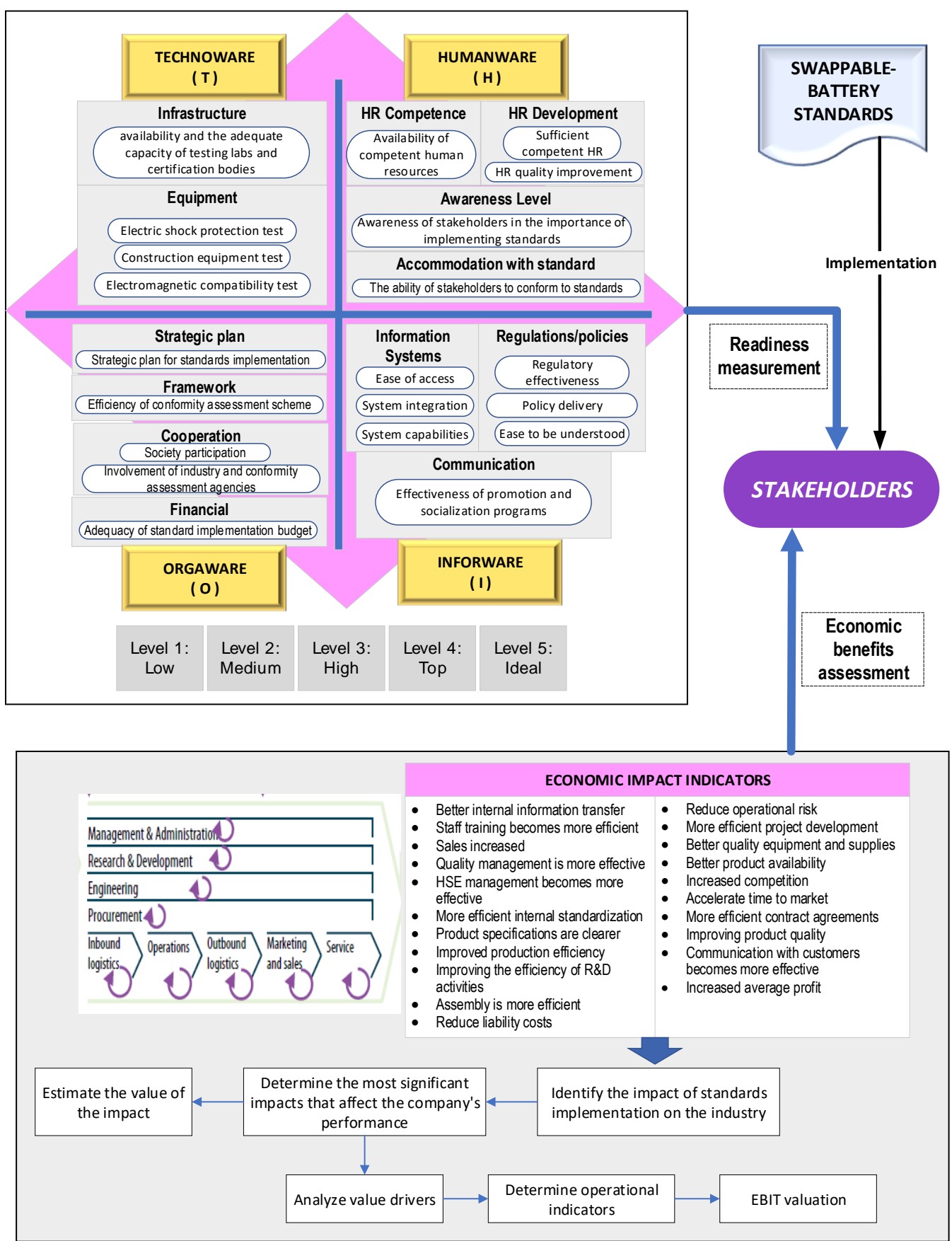

**Figure 4.** Model for measuring technology readiness and economic benefits of standard implementation.

### 3.2. Technology-Readiness Measurement

Based on the questionnaire recapitulation, data on readiness were obtained for technoware, humanware, inforware, and orgaware components. The data that were collected were processed using descriptive statistics to determine the percentage of achievement on each readiness scale. From here, it could be observed which indicators met the criteria and the extent of their achievements; this was then used as the basis for improvements to increase the readiness for these indicators.

### 3.2.1. Readiness Level

The first step in processing readiness data is looking at the readiness value for each question item or indicator by calculating the average score for each question item. From this calculation, the average value that shows the level of readiness of each item/indicator is obtained. Table 5 shows the data-processing results for the readiness value of each indicator.

**Table 5.** Readiness level of each indicator.

| | Indicators | Readiness Level |
|---|---|---|
| Technoware | The adequate capacity of available conformity-assessment agencies | Medium |
| | Test equipment for protection against direct electric contact | Medium |
| | Stored-energy test equipment | Medium |
| | Fault-protection test equipment | Medium |
| | Protective-conductor test equipment | Medium |
| | Battery-swap system construction test equipment | High |
| | Clearance and creepage-distance test equipment | Medium |
| | Material-strength test equipment | Medium |
| | Electromagnetic-compatibility test equipment | Medium |
| Humanware | HR competence | High |
| | Sufficient number of HR | High |
| | Effectiveness of coaching and training programs | High |
| | HR awareness of the importance of implementing standards for battery-swap system | High |
| | The ability of stakeholders to conform to standards | Top |
| Inforware | Easy access to information | High |
| | Optimal information-system integration | Medium |
| | Information-system capabilities | Medium |
| | Regulatory effectiveness | High |
| | Policies are well conveyed | High |
| | Policies are easy to understand | High |
| | Effectiveness of promotion and outreach programs | High |
| Orgaware | Availability of strategic plans | High |
| | Ease of obtaining permits, administration, and standards certification | High |
| | Society participation | High |
| | Involvement of industry and conformity-assessment agencies | High |
| | Financial adequacy | High |

Based on Table 5, the proportion of the readiness from each readiness scale to all the indicators can be calculated. The proportion was calculated by comparing the number of indicators at a certain level of readiness with the total number of indicators. Of the total

26 indicators, 38% were still in a medium state of readiness, 58% of the indicators had reached high readiness, and 4% of the overall indicators were at the top level. Meanwhile, there were no indicators at a low level, but none had reached the ideal readiness level. The proportions for each of these readiness levels are illustrated in Figure 5.

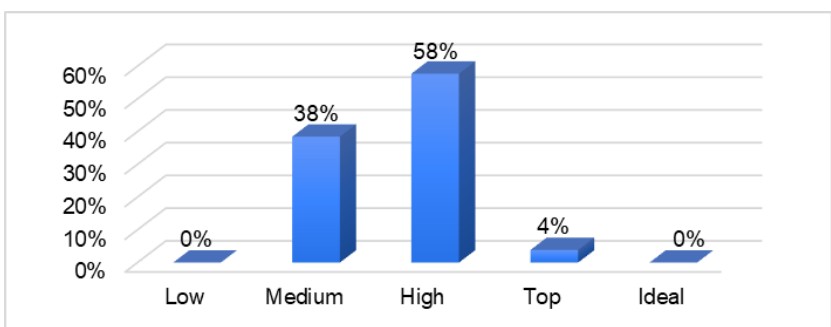

**Figure 5.** The proportions of readiness levels of all indicators.

3.2.2. Technology Contribution Coefficient (TCC)

The contribution of each component to the overall level of readiness could also be explored. Then, the TCC value could also be calculated. In this case, the value of each component's contribution was the level of readiness of each component. To determine each component's readiness level, it was necessary to refer to the results of the data processing in Table 5. As shown in Table 5, each component had a range of readiness values, meaning that it had an upper limit and a lower limit, describing the variation in conditions. To overcome this variation in conditions, the value of readiness used was the average score for each component, calculated by adding up the average values for all the indicators in the component and then dividing by the number of indicators in that component. Table 6 shows the data-processing results for the readiness value of each component.

**Table 6.** The data-processing results for the readiness value of each component.

|  | Average Value | Readiness Level |
|---|---|---|
| Technoware | 2 | Medium |
| Humanware | 3 | High |
| Inforware | 3 | High |
| Orgaware | 3 | High |

The next step is to calculate the TCC, which is a function of the technology component (THIO) and is formulated as follows:

$$TCC = \alpha \times T^{\beta_T} \times H^{\beta_H} \times I^{\beta_I} \times O^{\beta_O} \qquad (8)$$

$\beta$ is the weight of the composition of each component (THIO) with each other, where $\beta = 1$. In this study, it was assumed that each component had the same weight, so $\beta_T = \beta_H = \beta_I = \beta_O = 0.25$. At the same time, $\alpha$ is the trend factor of technology, which relates to the importance of measuring readiness in this study. In this study, the value was assumed to be $\alpha = 0.2$ as explained in Section 2. Thus, the TCC can be calculated as follows:

$$TCC = 0.2 \times \left( T^{0.25} \times H^{0.25} \times I^{0.25} \times O^{0.25} \right) = 0.2 \times \left( 2^{0.25} \times 3^{0.25} \times 3^{0.25} \times 3^{0.25} \right) = 0.54$$

The calculation results for each of the above components were used to create a radar diagram. The technoware component had a contribution value of 1.19, while the humanware, inforware, and orgaware components had the same contribution value of 1.32. Figure 6 illustrates the readiness level or component contribution to the radar diagram.

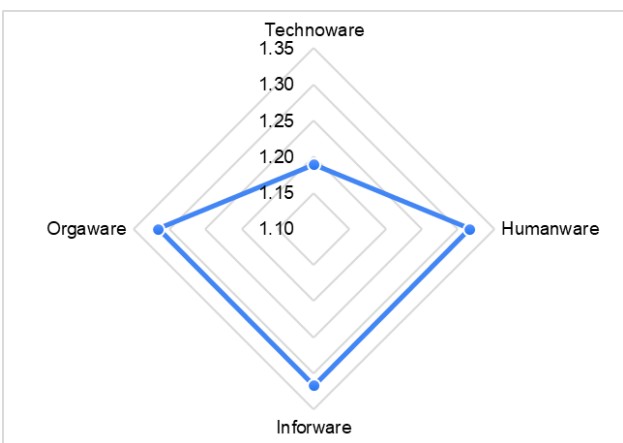

**Figure 6.** Radar diagram for the technology contribution coefficient.

### 3.3. Economic Benefits Assessment

Identification of Standard's Impacts, Significance, and Estimated Impact Value

Based on the recapitulation of the questionnaire, three kinds of data were obtained, which were included in the assessment of the economic benefits of the SB standard's implementation. The data collected were processed using descriptive statistics. The first data identify the impact of implementing the standard on the industry, representing respondents' responses to the 21 proposed impact indicators. Based on the results of the data processing, the answers or responses given by the respondents for all the indicators were "quite agree", "agree", and "strongly agree". The average score for each impact indicator could be calculated, showing the average response for each indicator. The average response for each impact indicator was "agree" and "strongly agree".

The second data identify the most significant impact that the battery industry will receive. From a total of twenty-one impact indicators, respondents were asked to choose five indicators that were the most significant or had the greatest impact. The data-processing results show the order of the most significant impacts based on the respondents' choices. A Pareto chart represents this sequence, as shown in Figure 7.

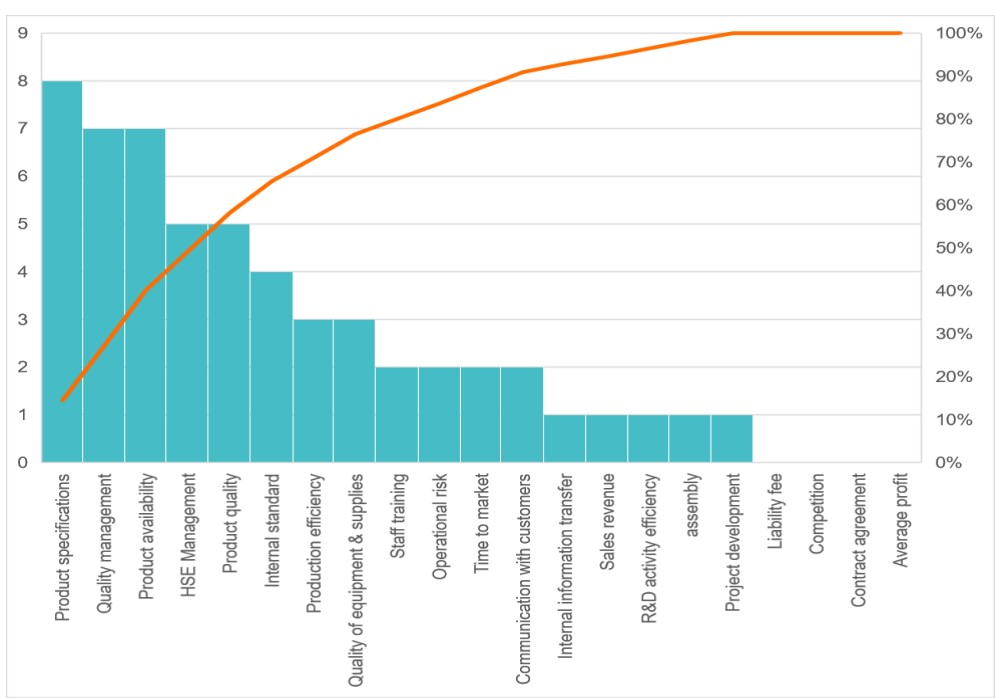

**Figure 7.** Pareto diagram of standard's most significant impacts.

The third data are the estimated impact values. These data were collected with the aim of assessing the percentage of benefits that would result from the related impact indicators when the standard was applied. The data were processed by calculating the average score given by 11 respondents. An average value of 4 was obtained based on the data processing, which means that the impact value was estimated to be 60–80%.

## 4. Analysis

The innovation of Electric Motorcycle Swap-Battery (EMSB) technology encourages the formation of a new ecosystem at the beginning of the supply chain, including technopreneurs and startups from manufacturers, suppliers, and distributors for commercialization. Swappable Batteries (SB), Electric Motorcycle (EM), and Battery Swap/Charging Station (BSCS) are the main components of the EMSB. In the EMSB system, motorcyclists exchange an electric motor battery that has run out of electricity for a fully charged battery. The Swappable Battery (SB) has a coupler for connecting the battery swap to an electric vehicle charger or to a charging rack, locking or unlocking the device, a battery management system, a temperature management system, an electrical protection circuit, and a battery-swap holder (International Electrotechnical Commission Technical Specifications 62840-1 -2016). The three main components of the EMSB have attracted the attention of supply-chain players and governments to find evolving solutions to enable faster adoption and diffusion of EMSB in Indonesia. In the future, the battery-swapping policy with interoperability standards will significantly reduce the cost of operation due to inter-brand collaboration and, therefore, will massively support commercialization for early investors. This is the reason to propose a swappable standard in the early stage.

The penetration of electric vehicles (EV) worldwide continues to progress rapidly due to technology improvements, urban air quality concerns, and rising consumer awareness. The EV market has grown significantly in many countries. There were approximately 190 million electric two-wheelers in the world at the end of 2020, which included electric motorcycles, mopeds, and scooters. Battery electric vehicles (BEV) accounted for 67.0 percent of global EV stock in 2020, while plug-in hybrid electric vehicles (PHEV) accounted for the remaining 32.7 percent [97]. Several experts and institutions predicted that the trend would continue in the coming years, accelerating the development of EV batteries. According to several studies, the increase in EV sales since 2009 is primarily due to government policy support during the early stages of EV market penetration [1–3].

Despite global EV stock growth, the country's trend with the world's most significant EV stock is undergoing several shifts. Prior to 2010, the world's two leading EV markets were the United Kingdom and the United States. Around 2010, Japan and Norway began to dominate the global EV stock, with 21% and 16%, respectively, while the United States maintained its lead at 22%. Since then, the United States had had the highest EV stock globally until 2016, when China surpassed it [24].

Other countries' progress in EV development and utilization varies. Several countries that appear to be ahead of the curve in EV development have financial incentives for EV customers and producers, accelerating the development of convenient and sufficient EV infrastructure. Southeast Asian countries, on the other hand, are just getting started with EV development. The differences and similarities in EV development progress and utilization in various countries, including Indonesia, are summarized in Table 7. Data are extracted and summarized from IEA reports for 2019-2021 [98–100] and various factual news sources.

**Table 7.** EV development in several countries.

| Country | Policy | | | Infrastructure | EV Market Share |
|---------|--------|--------|--------|----------------|-----------------|
| | Fiscal Incentives for EVs and Chargers | Hardware Standards for EV Chargers | National Action Plan | | |
| Norway | ✓ | ✓ | ✓ | around 16,000 charging points | 74.7% |
| China | ✓ | ✓ | ✓ | 976,000 public charging infrastructures [100] | 5.4% |
| US | ✓ | - | ✓ | >41,400 public normal charging points and 5000 fast charging infrastructures [100] | 2.2% |
| The Netherlands | ✓ | ✓ | ✓ | >50,000 of public charging infrastructure [101] | 24.6% |
| Germany | ✓ | ✓ | ✓ | >30,000 public normal and fast charging points [102] | 13.5% |
| UK | ✓ | ✓ | ✓ | 20,000 public normal and fast charging points [103] | 10.7% |
| India | ✓ | ✓ | - | 933 of public charging infrastructure [104] | 0.2% |
| Japan | ✓ | - | ✓ | around 18,000 public charging infrastructures [105] | 0.6% |
| South Korea | ✓ | - | ✓ | around 60,000 public charging infrastructures [106] | 0.4% |
| Thailand | ✓ | - | - | 1200 public charging infrastructures [105] | No record |
| Malaysia | - | - | ✓ | 300 public charging infrastructures [107] | No record |
| Indonesia | ✓ | - | ✓ | 122 public charging infrastructures [108] | No record |

The rapid development of EVs, particularly e-scooters and e-motorcycles, has led to the development of innovative electric scooter battery-swapping systems. Some of the pioneers who have successfully commercialized this innovation include Gogoro and Kymco from Taiwan. We have conducted a study on the development and implementation of business strategies for the commercialization of e-motorcycle and battery swapping technology and described the lessons learned [109,110]. Gogoro achieved above-average returns by emphasizing the value of innovation and overall lower cost than other electric motorbike companies; meanwhile, Kymco achieved above-average returns by emphasizing universal removable battery standards and extensive overseas networks and worldwide locations. In addition, the government regulations in Taiwan strongly support the development of the electric vehicle industry, so it encourages the industry to develop. In line with this, the Indonesian government has also released regulations and programs to support the development of EVs and battery-exchange systems in Indonesia. Specifically, Indonesia's National Energy General Plan sets a target of 2200 electric cars and 2.1 million electric motorcycles and the utilization of 2.3 TWh of electricity for electric vehicles by 2025. In 2019, the government issued a presidential regulation that aims to accelerate the use of electric vehicles, supplemented by derivative policies.

Furthermore, Maghfiroh et al. [23] explained that key stakeholders in Indonesia agree that EV technology has reached a high readiness level in technology development. However, available data show that Indonesia is still in the early stages of adopting electric vehicles [15]. More stringent policies are needed to provide an impetus for the development of electric vehicles.

### 4.1. Analysis of Level of Readiness to Implement SB Standard

This section contains the interpretation of the data-processing results for the condition of the readiness of EMSB stakeholders to implement the SB standard. The points analyzed in this section refer to the readiness criteria determined at the variable-operationalization stage. The discussion in this section is related to the readiness level for each criterion and the TCC value.

### 4.1.1. Technoware Readiness

Infrastructure is an essential factor for supporting the success of the standard application process. Based on the summary of the answers from the respondents, seventeen agencies have the potential to become testing labs, and eight agencies have the potential to become product-certification bodies. Several agencies can perform two functions (testing and certification) at once from these agencies. In addition, secondary data collection related to the population of testing labs and product-certification bodies in Indonesia was carried out; the data were obtained from the website of the national accreditation committee. From these population data, it is evident that there are 1322 testing labs and 605 accredited product-certification bodies. Each of these labs and institutions has a scope of testing and certification. Of the many institutions, only a few provide services for the battery scope. It is known that there are eight testing laboratories and nine product-certification bodies that provide services for battery scopes. However, the survey results show that the testing lab facilities and certification bodies in Indonesia adequately support the implementation of the SB standard.

The readiness of SB testing equipment is described as follows. Based on the readiness data, it is known that almost all the items in the technoware component have a "medium" readiness level, except for the "construction test equipment of swappable battery system" indicator, which has a "high" readiness level. The level shows that the testing equipment of conformity agencies in Indonesia is quite ready to test the battery. However, the readiness needs to be improved because, even though they have been able to test battery products, only a few agencies provide test services for electric vehicle applications.

### 4.1.2. Humanware Readiness

The human resource competencies referred to in this case are the capabilities, expertise, and skills of HR involved in implementing standards, standardization, and conformity assessments. This criterion ensures the availability of competent human resources for involvement in the implementation of standards. The survey shows that the human resources involved in implementing the SB standard have high competence. The competence is supported by the existence of various publications, research and development on electric vehicles, and discussions and forums that are often held by universities, manufacturing companies, and standards regulators.

Human resource development is an effort to improve the quality and quantity of human resources from various stakeholders involved in implementing standards. There is a high number of competent and qualified human resources from various stakeholders who play a role in implementing standards. The competency and qualification are supported by the high effectiveness of the coaching and training programs organized by the standardization body for human resource stakeholders in supporting the successful implementation of standards.

The level of awareness of various parties of the application of standards is an important aspect to be considered to support the successful implementation of standards. In this case, the stakeholders and public awareness level regarding the importance of implementing an SB standard also contribute to determining the level of readiness of human resources. Various parties involved in implementing the standard have a high awareness of the importance of implementing the SB standard.

The accommodation of the standard was analyzed with the aim of measuring how thriving stakeholders in Indonesia can comply with the SB standard when implemented. The accommodation of this standard could be through financial investment for meeting standard criteria, motivation, and a willingness to comply with standards. The survey shows that stakeholders in Indonesia are considered to adapt very well if standards are applied.

### 4.1.3. Inforware Readiness

The availability of a sound information system can support standard implementation activities by providing the information needed by various parties. The required infor-

mation includes standard documents, testing and calibration laboratories, certification bodies, standard accreditation and certification procedures, and much more. The national standardization body provides the information system for standardization and conformity-assessment activities, both online and offline, through the provision of Technical Service Offices spread across various provinces. The available information system has enabled good ease of access and has adequate integration and capabilities that are largely optimal in supporting the implementation of standards.

The existence of written regulations, procedures, and programs related to applying standards contributes to the readiness of information tools. The regulations referred to in this case are various regulations and programs related to electric vehicles and charging systems. Based on the survey results, regulations related to electric vehicles and charging systems support efforts to implement the SB standard, are well-conveyed to stakeholders and the wider community, create awareness, and educate stakeholders and the broader community regarding the importance of implementing standards.

In this case, the communication referred to is promotion and socialization to various parties related to standardization and conformity assessment. The standardization body has conducted promotion and socialization through standard dissemination to business actors, direct socialization by holding workshops with stakeholders, and promotion through social media such as Instagram and YouTube. This criterion was analyzed with the aim of measuring the effectiveness of the promotion and socialization programs that have been carried out. The survey shows that promotion and outreach programs for stakeholders and the public have effectively supported the successful implementation of the SB standard.

### 4.1.4. Orgaware Readiness

The strategic plan criterion aims to determine the extent of the availability of a strategic plan from the government and standards regulator to implement an SB standard. Based on the data-processing results, the indicator shows a high value of readiness, meaning that the strategic plan for implementing this standard has been well-structured.

The framework is a procedure for implementing standards through appropriate conformity-assessment activities. In this case, the framework used in standardization activities is a conformity-assessment scheme used as the basis for the harmonization of conformity-assessment procedures by conformity-assessment agencies. The conformity-assessment scheme was set efficiently, making it easier to manage permits, administration, and standard certification.

Collaboration with various parties to build public awareness and interest in applying standards is an essential form of cooperation to support the application of standards. The community actively participates in normal development activities to be easily implemented. In addition, industry and conformity-assessment agencies are actively involved in ensuring the infrastructure needed to implement the SB standard.

Financial support or funding from the government for implementing standards is a no-less-important factor to consider. This budget will later be used to provide infrastructure and equipment, develop standards, improve the quality of human resources, and run coaching and training programs. The state has sufficient finances to comply with standards and technology investments.

### 4.1.5. TCC Analysis

The TCC is the total contribution of the technological components that play a role in a system, taking into account the intensity of the contribution of each component. Obtaining the TCC value involved a series of assessments. The process involved statements/questions regarding various criteria and indicators to be prepared to implement the SB standard. The resulting TCC value represents the technological sophistication and technological readiness.

According to the calculations in the previous section, the TCC value is 0.54. Based on the classification of technological sophistication [111], the TCC value demonstrates a good

level of technological sophistication because it is between 0.5 and 0.7. The value shows that stakeholders in Indonesia are ready or able to implement an SB standard.

The magnitude of the TCC value is influenced by the magnitude of the contribution value of the components of technoware, humanware, inforware, and orgaware. Based on the radar diagram in Figure 4, the contribution value of each component is known. The technoware component has the lowest contribution value, which is 1.19, while the humanware, inforware, and orgaware components have the higher contribution value, which is 1.32. The difference in value is because the level of technoware readiness is lower than the levels of the other three components, where the readiness of technoware is still at the "medium" level. In comparison, the other three components have reached the "high" level. Technoware components need to be improved to increase the value of the TCC or the readiness of stakeholders as a whole. The technoware readiness can be improved by improving each of the criteria and indicators that have not yet reached the desired level.

### 4.2. Economic Benefits of Standard

4.2.1. Impacts of SB Standard Implementation

The potential impacts for the battery-manufacturing industry of the SB standard's implementation were identified. In addition, the most significant impacts were also identified based on the Pareto diagram. The five most significant impacts of the SB standard are as follows: clearer product specifications, more effective quality management, better product availability, more effective management of health/safety/environment (HSE), and product quality improvement.

Product specifications are essential in product development and product engineering activities. With the standard, companies do not need to develop product specifications independently. Product specifications that refer to standards can make it easier for the company to collect relevant materials, create product designs, and form processes. Thus, companies can save time and effort because they can directly adjust and refer to available standards. In addition, product design activities can be carried out more quickly because of the specifications already available in the standard. Thus, the time needed to develop a product is reduced, and the time to market can be shortened.

Quality management serves to maintain the level of quality desired by a company. It consists of quality control, quality assurance, and quality improvement. Quality management activities can be carried out more effectively based on standards when the latter exist. This can lead to a process of continuous improvement. When the overall quality is maintained, various failures in operations can be minimized, thereby reducing the possibility of rejects and defects being produced. Thus, the company can save many costs in the production process, which involves time, human resources, and materials, because it can minimize handling failed/defective products.

Product inventory control aims to determine the inventory, reduce the risk of delivery delays, and anticipate sudden changes in demand. The application of standards affects the warehousing needs, which will cause less inventory to be stored in the warehouse due to the high availability of standard products. The reduced need for warehousing will minimize the expense of managing inventory and the costs involved in keeping goods in the warehouse.

HSE management serves to protect workers and other people in the workplace by ensuring their safety and controlling risks from equipment, assets, and production sources, ensuring that they are used safely and efficiently to avoid accidents and occupational diseases. The effective implementation of HSE can also reduce various failure rates due to the risks posed by not using standard products and equipment. The number of accidents, injuries, and deaths can be reduced. In addition, improving the efficiency of HSE in the company can boost productivity in production activities, thereby reducing the consumption of energy in the form of fuel or electricity. Therefore, the existence of standards can reduce product failures that result in defective products and rejects, minimizing the number of products that must be discarded because they cannot be reused or repaired.

Improved product quality affects the responses of buyers or customers. Products that have been labeled standard must have passed a test and are suitable for use, so they can be trusted regarding the reliability of the features and specifications. Certified products will protect consumers, increasing buyer satisfaction with the products. Increasing the level of consumer satisfaction can reduce the number of complaints, such as those about the quality and functionality of the product. In addition, it can minimize the submission of warranty claims because the products purchased are of good quality. Thus, the company can save time that would be spent responding to customer complaints and reduce the probability of having to pay warranty compensation.

Based on the identification of the impact of implementing the above standards, a chart that explains the relationship between the significant impacts and the activities carried out for the existing business functions can be drawn, as illustrated in Figure 8.

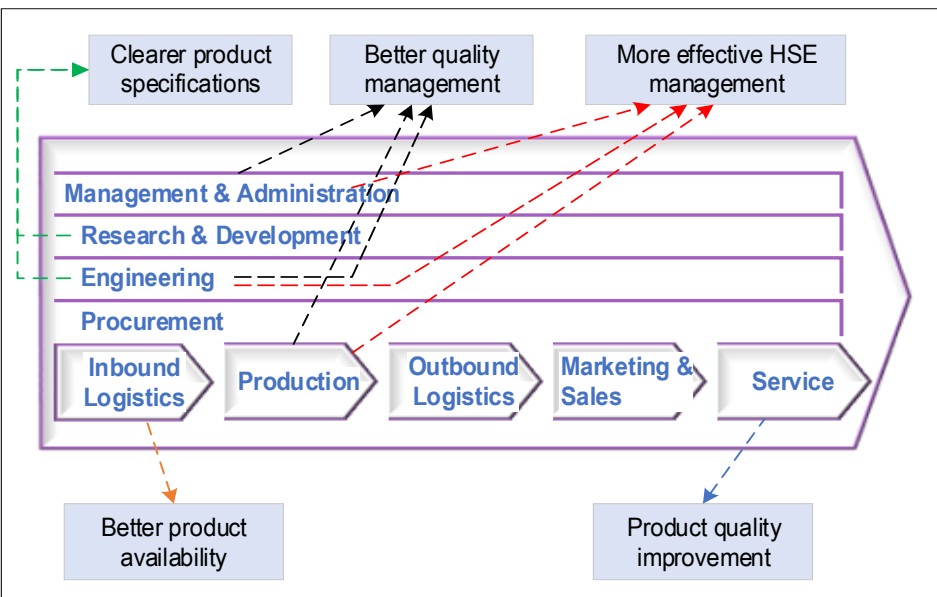

**Figure 8.** Impacts of SB standard on battery manufacturer's business functions.

### 4.2.2. Value Drivers and Operational Indicator Analysis

Value drivers are critical organizational capabilities that give companies a competitive advantage. The analysis of value drivers is crucial because it can help to elucidate the company's strategy and activities in various business functions that add value. If the impact of standards can be assessed for activities closely related to value drivers, their impact on value creation may be much higher than that in other activities. Value drivers can be related to the activities of a particular business function. They may extend to the activities performed by several business functions or even to specific operations of the entire company. Based on the impact identification described in the previous section, the business functions for which the impact provides added value could be observed. Then, the value drivers could be analyzed.

Operational indicators are measurable variables of company activities that show an increase or decrease in performance. Operational indicators are used to measure the impact of the standard on the activities performed by the selected business function and to measure its contribution to the creation of corporate value (contribution to the EBIT or gross profit of the company). The selection of appropriate operational indicators is one of the essential tasks of an economic benefits assessment. Concerning the significant impacts mentioned above, each has different operational indicators. Based on the impact identification in the previous section, operational indicators could be developed for each impact. Table 8 below shows the relationship between the impact of the standard, the associated value drivers, and operational indicators.

**Table 8.** Value drivers and operational indicators.

| SB Standard Impact | Value Drivers | Operational Indicators |
|---|---|---|
| Clearer product specifications | • Excellence in new product design<br>• R&D efficiency and effectiveness | • Labor cost savings<br>• R&D cost savings<br>• Time-to-market savings<br>• Increased demand |
| Better quality management | • Product quality<br>• Production process quality<br>• Production efficiency<br>• Continuous improvement | • Reducing costs in handling rejects, rework, and repair of defective products<br>• Production cost savings |
| Better product availability | • Product quality | • Reduction in inventory cost<br>• Reduction in warehousing costs |
| More effective HSE management | • Compliance with safety | • Reduced work accident insurance costs<br>• Waste-handling cost savings<br>• Energy cost savings per unit production volume |
| Product quality improvement | • Service quality<br>• Product quality | • Save time in handling customer complaints<br>• Cost reduction for warranty compensation payments |

### 4.2.3. EBIT

The earnings before interest and tax (EBIT) is a crucial indicator applied to valuation and used to measure company value creation. EBIT represents a company's gross profit, i.e., revenue minus expenses, at a given point in time. The implementation of the standard is expected to cause a change in the values of the operational indicators for the selected business function. This impact, converted into monetary units, shows that the value created by the firm is increased by (a) a reduction in costs, (b) a contribution to higher revenues, or (c) a combination of both. Figure 9 highlights the relationship between the value drivers and operational indicators determined, which are then summarized regarding the overall contribution to a company's EBIT. The aggregation of all these operational indicators produces the EBIT. This EBIT is the value of the economic benefits of implementing the SB standard in this study.

Based on the estimated impact on data processing, the percentage value is 60–80%. This value is an estimate of the change in a particular activity after implementing the standard. This value was used in the calculation of the EBIT. The revenue and expense elements shown in Figure 8 were also used to calculate the EBIT. To obtain the EBIT, we aggregated the estimated increase in the sales of battery packs and the estimated cost savings, including overhead costs, lithium battery production costs, and labor costs. Expenses, such as depreciation or capital expenditures, research and development, marketing, transportation and distribution, warranties, profits, and others, were included in the overhead costs. The elements of income and costs were estimated annually and multiplied by the percentage estimated impact value. Figure 10 shows the projected total economic benefit (EBIT) earned in one year after the battery-swap standard is implemented.

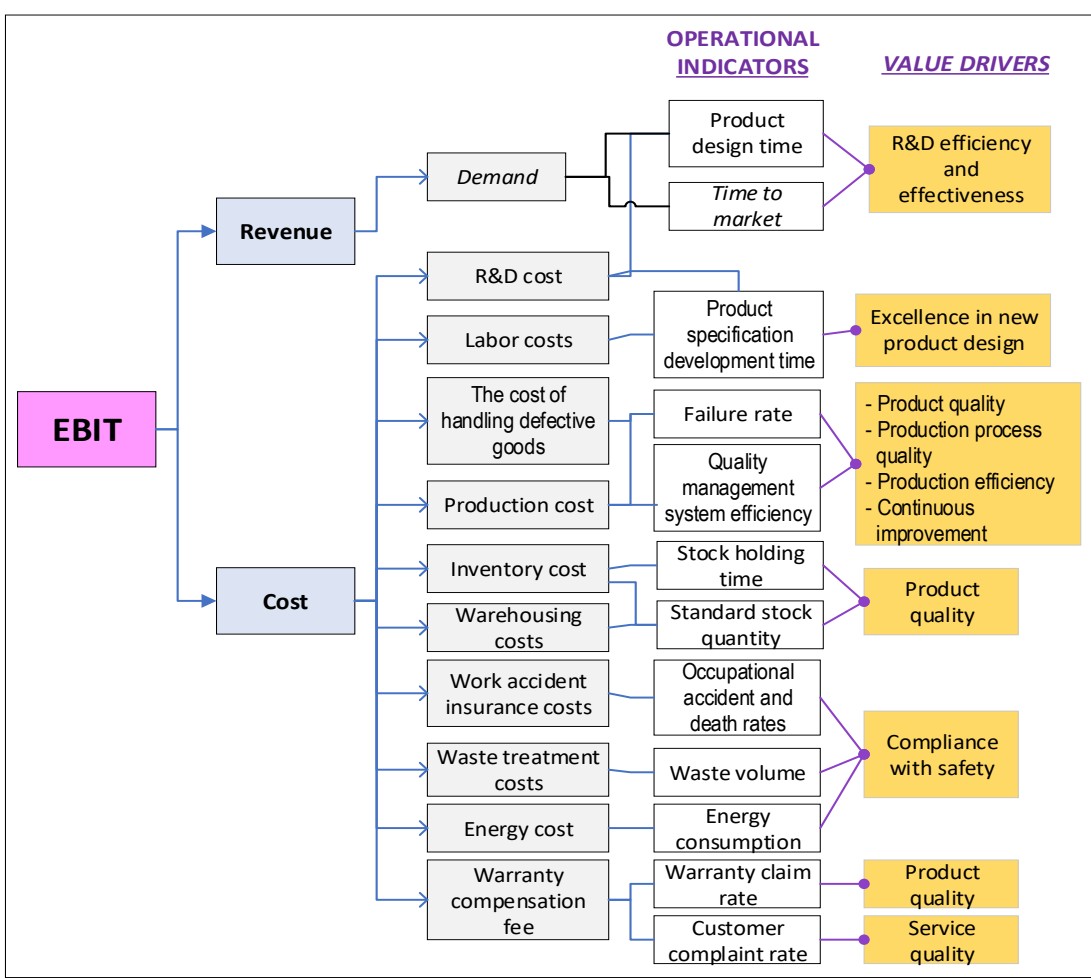

**Figure 9.** Relationship of EBIT, operational indicators, and value drivers.

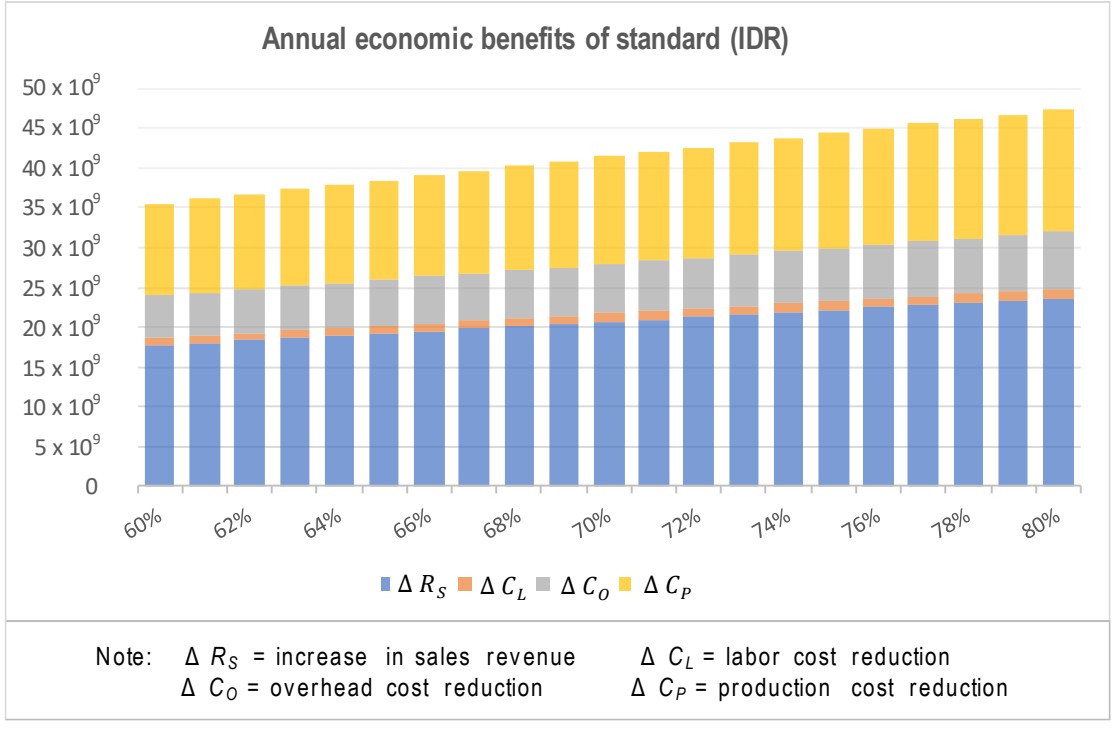

**Figure 10.** Projected economic benefits of swappable battery standard.

### 4.3. Benefit–Cost Ratio

We conducted a feasibility study of implementing a standard battery swap by comparing the overall costs and benefits. The costs component was generalized based on the measured readiness level. Meanwhile, the benefits component was compiled based on the impact identification and the impacts' estimated value according to the analysis.

This study's generalization of the cost components refers to the cost model, which classifies costs into three categories: capital expenditure, implementation costs, and training costs [56,96]. In this study, the capital expenditure is the cost incurred by the need to repair technoware components, which involves infrastructure and testing equipment. Based on the analysis of the level of readiness, it is known that it is necessary to make improvements to the technoware component to improve its readiness. The implementation costs are defined as the costs of implementing the standard; this cost is an aggregation of all the costs that must be incurred by the stakeholders involved. These costs were generalized based on readiness indicators for the humanware, information, and orgaware components. The costs related to training include bringing technology experts to train employees, training materials, expert fees, and costs for employees who attend training. The following Table 9 shows a recapitulation of the cost components that have been processed.

**Table 9.** Cost components.

| Type of Cost | Nominal (IDR) | Source |
|---|---|---|
| Accreditation fee for additional testing scope and swap−battery system certification ($C_{T_1}$) | 472,500,000 | Swappable battery standard [50] and Government Regulation No. 40 year 2018 |
| Procurement of testing tools, machines, and software ($C_{T_2}$, $C_{T_3}$, $C_{T_4}$) | 790,275,000 | Data processing |
| HR training and coaching ($C_{H_1}$) | 2,005,000,000 | |
| Quality improvement of the Technical Service Office ($C_{I_1}$) | 2,700,000,000 | |
| Information system improvement ($C_{I_2}$) | 3,000,000,000 | |
| Policy development ($C_{I_3}$) | 2,000,000,000 | National Standardization Body strategic plan [112] |
| Standard development ($C_{I_4}$) | 800,000,000 | |
| Counseling, workshops, seminars, and dissemination related to standards ($C_{I_5}$) | 400,000,000 | |
| Development of conformity assessment scheme for battery-swap system ($C_{O_1}$) | 46,875,000 | |
| Certification ($C_{O_2}$) | 40,000,000 | Government Regulation No. 63 year 2007 |
| Tesing ($C_{O_3}$) | 8,812,000 | Regulation of the Minister of Finance of the Republic of Indonesia No. 214/PMK. 05/2020 |
| Total cost (C) | 12,263,462,000 | Data processing |
| Operation and maintenance (O&M) | 20,000,000 | Data processing |
| Salvage value (SV) | 158,055,000 | Data processing |

The generalization of the benefits component was obtained from the benefits assessment analysis that was carried out, where the most significant impacts were identified. From the analysis carried out for the assessment of the economic benefits, it is known that the operational indicators measure the value of the impact caused and can be expressed as an entity that can be quantified, in the form of either cost savings or increased revenues. The aggregate of all the operational indicators is expressed in the EBIT and represents the value of the economic benefits that the company will obtain. Therefore, in this case, the

EBIT value was included in the benefits component with a deduction of the applicable income tax.

- Increase in sales (Bs)

Assuming one electric motorcycle requires one battery pack, then total sales of electric motorcycles = sales of battery packs. Demand for electric vehicles in the first year = 7601 units [113]. Battery capacity for one motor unit = 2 kWh (e-viar.com/order;2021 [113]). Thus, the demand for battery packs = 7601 units/year × 2 kWh = 15,202 kWh/year, and the calculation is summarized in Table 10.

**Table 10.** Calculation of the increase in sales.

|  | **Nominal** | **Unit** | **Source** |
|---|---|---|---|
| Demand | 15,202 | kWh/year | [113] |
| Selling price | 137 | $/kWh | [97] |
| Sales | 2,082,674 | $/year | |
| Benefit (60%) | 17,744,382,480 | IDR/year | |

- Labor cost reduction ($B_l$)

The labor cost for lithium battery production is assumed to be 5% of the selling price of the battery [114]. Thus, labor costs can be calculated as follows:

Labor cost in 1 year = 5% × selling price × demand = 5% × $137/kWh × 15,202 kWh/year = 104.133.70 $/year = IDR 1,478,698,540/year.

Benefit of reducing labor costs = 60% × IDR 1,478,698,540 = IDR 887,219,124 per year.

- Overhead cost reduction

Costs such as depreciation or capital expenditures, research and development (R&D), marketing, transportation and distribution, warranties, profits and others are included in overhead costs [97]. So, in this case, overhead costs consist of R&D costs ($C_r$), costs for handling defective/failed items ($C_f$), inventory costs ($C_i$), warehousing costs ($C_w$), work accident insurance costs ($C_{ai}$), waste-handling costs ($C_{wm}$), and warranty compensation fee ($C_{wc}$). Then, the overhead cost for lithium battery production is assumed to be 30% of the selling price [114]. Thus, the overhead costs can be calculated as follows:

Overhead cost = 30% × selling price × demand = 30% × $137/kWh × 15,202 kWh/year = 624,802.20 $/year = IDR 8,872,191,240/year.

Benefit of reducing overhead costs = 60% × IDR 8,872,191,240 = IDR 5,323,314,744 per year.

- Production cost savings ($B_p$)

Based on the research of Patry et al. [114], the production cost of lithium batteries is 65% of the selling price. Thus, the production cost can be calculated as follows:

Production cost = 65% × selling price x demand = 65% × $137/kWh × 15,202 kWh/year = 1,353,738.10 $/year = IDR 19,223,081,020/year.

Production cost reduction benefit = 60% × IDR 19,223,081,020/year = IDR 11,533,848,612/year.

Based on the calculation of the benefits that have been carried out, a recapitulation of the entire benefits component can be arranged as summarized in Table 11:

**Table 11.** Benefits component recapitulation.

| **Type of Benefit** | **Nominal (IDR)** |
|---|---|
| Increase in sales | 17,744,382,480 |
| Labor cost reduction | 887,219,124 |
| Overhead cost reduction | 5,323,314,744 |
| Production cost savings | 11,533,848,612 |
| **Total economic benefits (EBIT)** | **35,488,764,960** |

From the EBIT value obtained, it is then deducted by the tax burden to obtain the EAT (Earnings After Tax) value or profit after tax. The income tax rate for corporate entities is 25% of taxable income (Law Number 36 of 2008). Thus, the EAT value can be calculated as follows:

Income Tax Expense = 25% × EBIT = 25% × IDR 35,488,764,960 = IDR 8,872,191,240.

EAT = EBIT—Income Tax Expense = IDR 35,488,764,960—IDR 8,872,191,240 = IDR 26,616,573,720.

Thus, the benefit value that will be considered in the calculation of the B/C ratio is the EAT with a value of IDR 26,616,573,720.

As shown in Figure 11, the total benefit of implementing a standard battery swap is eight times greater than the total cost of increasing stakeholder readiness. Thus, standard implementation is feasible.

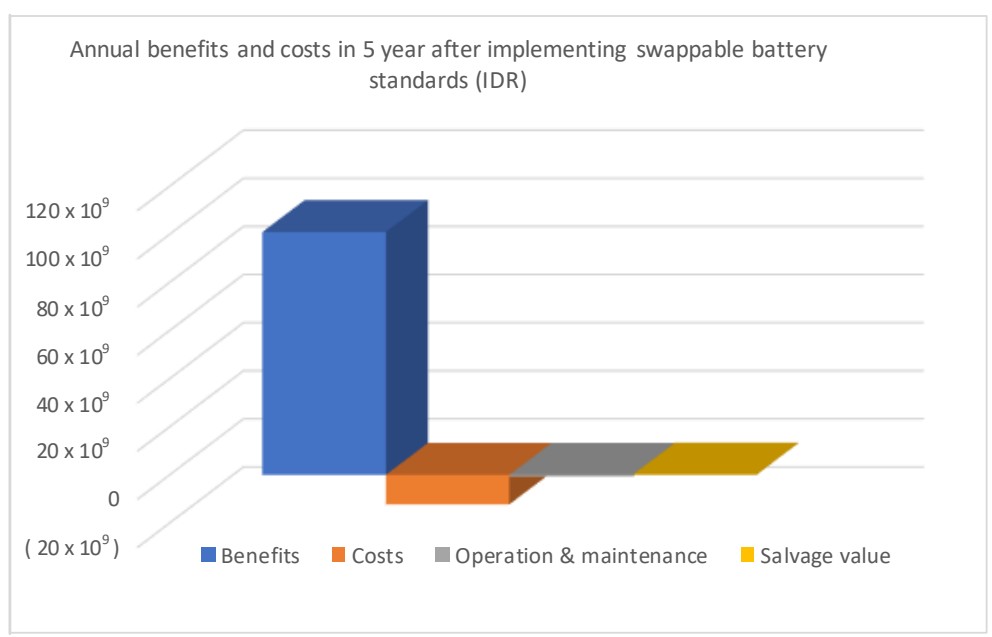

**Figure 11.** Annual benefits and costs in 5 years after implementing SB standard.

### 4.4. Model Applicability

A model for measuring the readiness of stakeholders in implementing the SB standard was successfully established, and this model resulted in the identification of the economic benefits of implementing a swap-battery standard for the battery industry. Thus, this measurement framework or model can be used to evaluate the readiness and ability of stakeholders to implement a standard so that, from the evaluation results, strategies and recommendations can be formulated to improve the existing conditions. The obtained model outputs can prove the research hypothesis. The first output is in the form of a readiness value for each criterion for each technology component (technoware, humanware, inforware, and orgaware), which shows the ability of stakeholders to meet the readiness criteria for implementing the standard swap-battery test system. In addition, the value of the technology contribution coefficient (TCC) represents the technological sophistication, technological readiness, and capability of the existing system being measured.

The output of the economic benefits assessment is the identification of the potential impacts leading to significant positive economic changes for the battery industry. In addition, the model also generated an output of an estimate of how significant the value of the change is so that the impact of implementing the standard could be quantified; then, the EBIT was obtained, which is the aggregation of all the operational indicators for each impact of the standard's implementation. The EBIT is the value of the economic benefits of implementing the standard.

Furthermore, the developed model also provides an output in the form of an analysis of the feasibility of implementing the standard, expressed as the benefit–cost ratio (B/C ratio). The B/C ratio compares the economic benefits derived from implementing the standard with the overall expenditure required to improve or increase readiness. Based on the benefit–cost ratio analysis in the case study of this research, it is known that the SB standard's implementation is feasible.

The developed measurement model also integrated the standard application approach, which considers aspects of technology adoption and economic benefits in a single unit simultaneously. Adopting technology in this study means that the existing system observed (EMSB stakeholders) adopts a new entity (standard) to achieve specific goals, namely, improving the quality and competitiveness of domestic products, protecting producers and consumers, and shortening the time to market for product innovation (battery swap). Thus, the model can be used as a benchmark to evaluate stakeholders' ability and readiness to implement the SB standard.

### 4.5. Proposed Recommendations for EMSB Stakeholders

#### 4.5.1. Recommendations for Standard Regulators

The analysis shows a high level of awareness of various parties, especially the battery and electric vehicle manufacturing industry, regarding the importance of implementing an SB standard. Therefore, it is essential for standards regulators to immediately promote these standards in Indonesia by adopting and implementing standards adapted to the country's character, needs, and actual conditions. The adaptation needs to pay attention to domestic capabilities to adapt more quickly to the standards adopted and applied. Thus, consumer and producer protection can be realized more quickly, and the competitiveness of domestic products will be better. Therefore, standards regulators need to map the needs of battery-swap stakeholders and the ability of stakeholders to refer to the standard.

#### 4.5.2. Recommendations for Product Testing and Certification Agencies

Based on the analysis of the TCC value, stakeholders in Indonesia are ready to implement an SB standard. However, the technoware component (technical tools) needs to be improved if one wishes to increase the value of the TCC or the readiness of stakeholders as a whole. The technoware readiness can be improved by improving product testing and certification service providers. In this case, it is necessary to look at the adequacy/availability of these institutions, infrastructure readiness, and capacity.

The analysis shows that few institutions are expected to provide services for the battery scope. To anticipate the surge in demand for swap-battery testing and certification services in the future, institutions that can perform this function need to expand their scope of testing. The expansion of the scope of testing will affect the provision of the necessary infrastructure and equipment. In addition, to support the commercialization of this swap-battery technology, these institutions need to increase their capacity to carry out tests for large capacities. These aspects can be achieved by investing in appropriate equipment and infrastructure procurement. In this study, the details of the infrastructure procurement based on the expansion of the required testing scope have been described. This research can be used as a reference or illustration for testing labs and product-certification bodies in Indonesia for improving infrastructure readiness. Thus, this research can provide practical benefits in an overview of the investment before the SB standard is implemented.

#### 4.5.3. Recommendations for the Battery-Manufacturing Industry

The analysis shows that implementing the standard will positively impact the battery industry, evidenced by the considerable economic benefits obtained, especially through cost savings in most business chains. The application of standards will provide benefits if the industry can utilize them optimally as a driver of product excellence. As Indonesia's lithium battery and electric vehicle businesses are still in the growth phase, the selling price offered is still likely to be expensive for consumers (the B2C business model). The business

model will probably cause penetration in the market to take a long time. Therefore, in promoting product excellence, the business model can be shifted from B2C to B2B (business to business). For example, the battery industry supplies its battery-pack products to the electric motorcycle industry to be installed on the motorcycle body. In addition, another step that can be taken is to collaborate with other industry players through the B2B2C (business to business and business to consumer) business model scheme. For example, battery manufacturers supply battery-pack products to the electric motorcycle industry to provide battery-exchange stations for consumers.

Product certification with standard labels is one of the drivers of product excellence. In the future, if the battery-swap standard has been applied systematically and the battery industry has obtained product certification, product certification must be used optimally as a means of promotion to introduce product advantages and as a means of guaranteeing the quality of a product for consumers. A standard certificate marking that is clear and can be directly seen by customers will provide convenience and distinguish it from other products. Certification makes it easier to negotiate with counterparts or customers who have ISO certification because they will look for suppliers in the same class or that are familiar with the standard. Promotion and marketing teams will become more confident in the promotion process because they will have reliable weapons for global trade competition.

### 4.5.4. Recommendations for the Government

This research can be recommended to the government to be used as part of a road map to help to accelerate the battery-based electric vehicle program through a study of the application of battery-swap standards. In this case, the government needs to map the required test equipment, map the industrial production capacity, and test the infrastructure capacity. To expedite and simplify the application of standards, the government needs to coordinate with various testing labs and certification bodies that have been identified, battery and electric vehicle manufacturing industries, standard regulators, and consumers. The coordination is necessary to develop programs and strategic plans that are more comprehensive and accurate so that all the parties are not burdened by the implementation of this standard and can receive positive impacts from the implementation of the standard.

### 5. Discussion: Open Innovation of Swappable Battery Standard

In open innovation, external and internal ideas and internal and external paths to market are used to advance new technology [115]. In this process, new sources of knowledge are explored to foster innovation opportunities for existing products and intellectual property rights (IPR). In addition, collaboration with various parties, such as customers, government, academics, and companies, is also carried out to explore the potential for technology development. The process of open innovation is correlated with the principles applied to standards development, namely, transparency, openness, and considering the needs of various stakeholders. Based on the concept of open innovation of Chesbrough [116], Gassmann and Enkel [117], and the principle of standards development, open innovation in swappable battery standardization can be defined into three stages: inbound process, outbound process, and interaction process.

The inbound process includes the integration of scientific and technological developments and the external knowledge of experts representing various stakeholders from a variety of organizations. This process aims to gather stakeholder perspectives to ensure common understanding and address the interested parties' needs. The development of swappable battery standards in Indonesia has been carried out for several years involving academia, manufacturers, consumers, government, and a standardization agency [118–122].

The outbound process is a process to provide standards users with standardization outcomes, such as standards, policies, and regulations, that meet the needs of relevant stakeholders. The standards are introduced into the market and business environment in this process. The standardization agency has also carried out this stage through standard

dissemination to battery manufacturers, direct socialization by holding workshops with stakeholders, and promotion and outreach to various related parties.

The interaction process is where the new standards are developed based on previous versions, and new participants are introduced in the standards development process. Thus, standards users can become standards developers. In this stage, stakeholders involved in developing swappable battery standards have learned from previous knowledge and added new experiences and developments to prepare new standardization outcomes.

The dynamics of open innovation on the swappable battery standard in Indonesia can be illustrated through previous studies that show developments in technology and standardization. These include the development of formulas and critical production technologies [123–127], standards related to testing and product quality that serve as a reference for the preparation of Indonesian national standards [89,118,119,121,122,128,129], testing entity strengthening models [66,130], and the development of cost estimation models for lithium battery cells, modules, and packs [131,132]. We provide an early standardization scheme in Figure 12 to describe the dynamics of open innovation in swappable battery standardization.

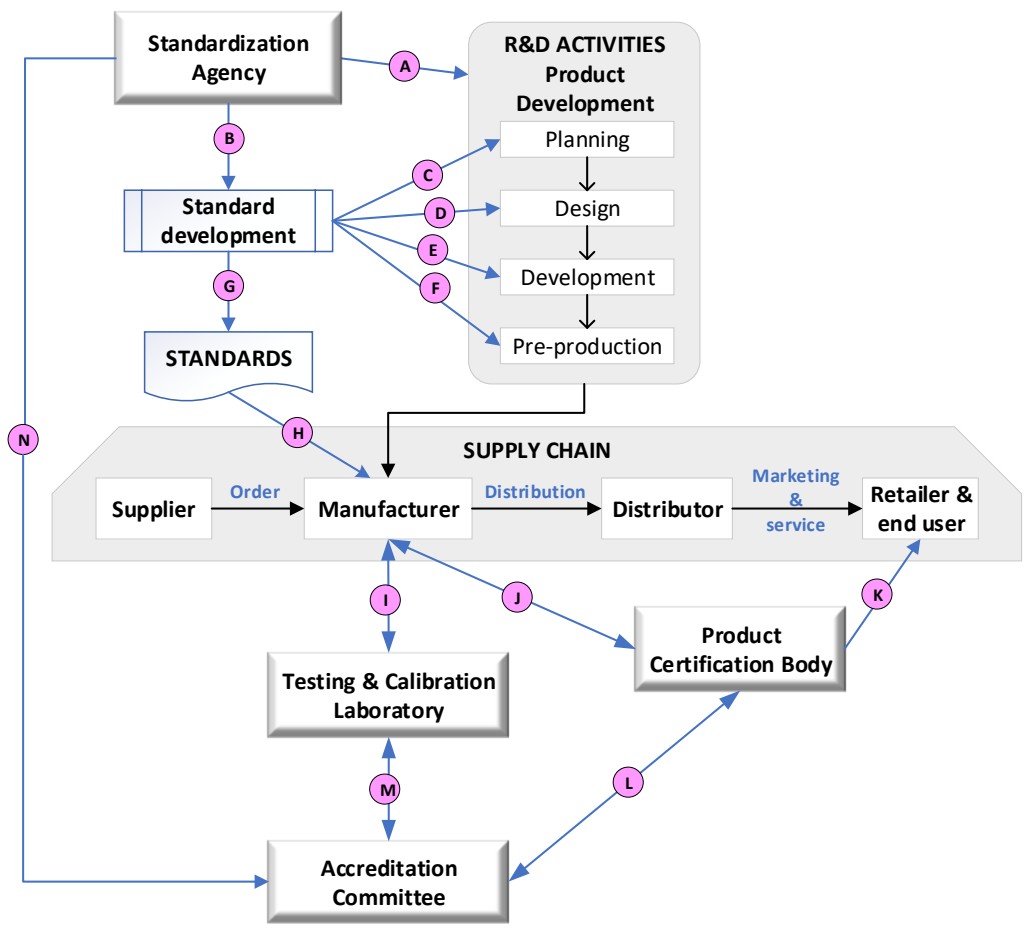

**Figure 12.** Early standardization scheme.

Overall, five stakeholders play a role in the early standardization scheme. The first stakeholder is the Standardization Agency as the body that provides training, development, and coordination of activities in standardization, as shown in arrows A and B. Then, the Manufacturer is the next stakeholder as the center for battery production. As indicated in the arrow I, the Testing and Calibration Laboratory is the institution conducting conformity assessments of manufactured products. In addition, the product certification agency is in charge of certifying battery products, as shown in arrow J. The Accreditation Committee

is also involved in this standardization activity, executing the duties and functions of the Standardization Agency in the field of accreditation as shown in arrow N.

Before battery products are mass-produced and used for electric vehicle propulsion, developing testing standards for the swappable battery is necessary. The standard development process is the responsibility of the Standardization Agency as a supervisor, developer, and coordinator of activities in the field of standardization. The development of standards at the beginning needs to pay attention to the initial supply chain of the swappable battery itself, namely, at the product development stage, which includes planning, design, development, and pre-production. As shown in arrows C, D, E, and F, the formulation and set of standard activities are carried out in each product development process. After the standard has been formulated and established, it is implemented in the mass-production process at the battery Manufacturer, as indicated by arrow H.

Swappable battery products can be certified if Testing and Calibration Laboratories and Product Certification Bodies certify that the product is suitable for consumer use. As shown in arrow K, the recommended certification scheme requires products circulated to be retested [133]; therefore, users' feedback is needed. Furthermore, requirements for establishing Test and Calibration Laboratories and Product Certification Bodies must follow the accreditation requirements set by the Accreditation Committee, as shown in arrows L and M.

## 6. Conclusions

### 6.1. Implications

This paper constructs a model of measuring technology readiness and economic benefits in the open innovation of a swappable battery standard. This model employs a technometrics analysis to measure the stakeholders' performance in the context of technology readiness. In addition, the ISO methodology assesses the economic benefits of the swappable battery standard. The measurement framework in this paper can provide another insight to analyze the performance of the standard implementation.

The implication of the TCC result in the technometrics analysis is to drive the readiness level of each technology component (THIO) of implementing the swappable battery standard. It also provides valuable information for improving the standard implementation and stakeholders' performance. In detail, a policy brief is provided to propose recommendations for stakeholders on navigating within the open innovation ecosystem of the swappable battery standard in Indonesia.

The economic benefits assessment result can provide insight and understanding to stakeholders that the implementation of standards has a positive impact on the industrial economy, which increases the nation's competitiveness. Thus, this research is expected to raise awareness of the importance of adopting and implementing swappable battery standards and improve the enthusiasm of stakeholders to be involved in the open innovation ecosystem of standardization.

Theoretically, this research integrates a standard implementation concept approach that simultaneously considers aspects of technology adoption and economic benefits. This research supplements the literature on EV standardization in Indonesia, technology-readiness measurement, and the economic benefits of a standard. This research is novel in providing an overview of the readiness of stakeholders in Indonesia and the economic benefits of implementing the swappable battery standard, a summary of investments before the standard is implemented, and proposed recommendations for the stakeholders involved.

The proposed model and measurement tool can assist in the decision-making process. The decision making can be administered through the output of measurement results in the form of the achievement level of a system to provide consideration to stakeholders. In addition, the decision-making process can also be assisted by the presence of strategies, policies, and recommendations that are synthesized based on the application of measuring tools. Likewise, in this study, the resulting model was used to make recommendations for the stakeholders involved in implementing the standard.

*6.2. Limits and Future Research Topic*

The measurement model can be implemented and further evaluated to measure the readiness, open innovation, and economic benefits when stakeholders have implemented the swappable battery standard. Further research can generate a comparison between before and after standard implementation. Thus, the value of the economic benefits can be calculated with certainty, and feasibility studies can be carried out accurately and comprehensively. In addition, it is necessary to conduct a direct survey of the battery industries in Indonesia to procure production, financial, and other data so that the benefit–cost ratio can be calculated more accurately. In implementing the model for different standards, it is necessary to pay attention to the testing parameters required by these standards, the provisions of the respondents, the use of measuring instruments, and the party carrying out the measurements so that the measurements are right on target and adequately implemented. Furthermore, some of the limitations of this study may present risks inherent in the proposed policy recommendations. Risk management needs to be followed up to classify what risks are contained, especially in calculating the investment that has been described and the estimated value of economic benefits.

**Author Contributions:** Conceptualization, E.F.A., W.S. and E.P.; methodology, E.F.A., W.S. and E.P.; software, E.F.A.; validation, W.S., E.P. and A.M.; formal analysis, E.F.A., F.F. and M.H.; investigation, E.P., M.H., F.F. and A.M.; resources, W.S., F.F. and M.H.; data curation, E.F.A.; writing—original draft preparation, E.F.A.; writing—review and editing, E.F.A. and W.S.; visualization, E.P. and A.M.; supervision, W.S. and E.P.; project administration, E.F.A. and W.S.; funding acquisition, W.S. and M.H. All authors have read and agreed to the published version of the manuscript.

**Funding:** This research was funded by the Institution of Research and Community Services, Universitas Sebelas Maret, through the program "Penelitian Kolaborasi Internasional", grant number 260/UN27.22/HK.07.00/2021.

**Institutional Review Board Statement:** Not applicable.

**Informed Consent Statement:** Not applicable.

**Data Availability Statement:** Not applicable.

**Acknowledgments:** The authors would like to thank "Badan Standardisasi Nasional (BSN)" for supporting this research.

**Conflicts of Interest:** The authors declare no conflict of interest.

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
