# Peer review of "Technology Readiness and Economic Benefits of Swappable Battery Standard: Its Implication for Open Innovation"

_2199-8531, doi:10.3390/joitmc8020088_

Round 1
Reviewer 1 Report
The author has address all the concerns and issues.
Thank you for the great effort
Author Response
Surakarta, April 29, 2022
Dear Academic editor
We would like to inform you that we will re-submit my Manuscript Manuscript ID: JOItmC-1698398, Type of manuscript: Article; Title: Model for Measuring Technology Readiness, Open Innovation, and Economic Benefits of Swappable-Battery Standard for Electric Motorcycles in
Indonesia. We have already changed to a New Title based on Academic Editor: “Technology Readiness and Economic Benefits of Swappable-Battery Standard: its implication for open innovation."
We also give a list of improvements in a table based on Academic Editor’ comments and the changes already indicated by highlighted red i.e. add with more than 10 papers on open innovation dynamics (non JOI papers); add "open innovation'" as keywords; and changed ‘Discussion’ with ‘Analysis’, and added section 6. In conclusion, with 6.1. Implication (theoretical implication, practical implication); and 6.2. Limits and Future Research Topic (add)
In this improvement paper, we fill more details about the gap in the research on “Technology Readiness and Economic Benefits of Swappable-Battery Standard: its implication for open innovation ."There has been no research that has measured technology readiness for adopting standards while, at the same time, assessing their economic benefits and vice versa. Therefore, a model for measuring technology readiness for adopting a standard and, at the same time, evaluating the economic benefits of implementing the standard is needed to accelerate the technology commercialization process. This paper discussed the innovation of electric-motorcycle swap-battery (EMSB) technology. This study aimed to propose a model with which to measure the technology readiness of the EMSB's stakeholders in implementing the swappable battery (SB) standard. We developed the technometric framework and Economic Benefits of Standards - ISO Methodology 2.0. We believe that this paper deserves to be published in the Journal of Open Innovation: Technology, Market, and Complexity, because it discusses new topics and give a novel in providing an overview of the readiness of stakeholders in Indonesia and the economic benefits of implementing an SB standard, an overview of investments before the standard is implemented, and proposed recommendations for the stakeholders involved.
We declare that there is no conflict of interest to disclose. Please address all correspondence regarding this manuscript to wahyudisutopo@staff.uns.ac.id. We are looking forward to your response according to this submission.
Best regards
Era Febriana Aqidawati, Wahyudi Sutopo*, Eko Pujiyanto, Muhammad Hisjam, Fakhrina Fahma, and Azanizawati Ma’aram

Reviewer 2 Report
The paper has been organized and written well.
Author Response

(The authors gave the same response as above.)
